# Machine Learning in Predictive Maintenance towards Sustainable Smart Manufacturing in Industry 4.0

**Zeki Murat Çınar [1], Abubakar Abdussalam Nuhu [1], Qasim Zeeshan [1,*], Orhan Korhan [2], Mohammed Asmael [1] and Babak Safaei [1,*]**

[1] Department of Mechanical Engineering, Eastern Mediterranean University, Famagusta 99628, North Cyprus via Mersin, Turkey; cinar.zekimurat@gmail.com (Z.M.Ç.); abubakar.nuhu@emu.edu.tr (A.A.N.); mohammed.asmael@emu.edu.tr (M.A.)

[2] Department of Industrial Engineering, Eastern Mediterranean University, Famagusta 99628, North Cyprus via Mersin, Turkey; orhan.korhan@emu.edu.tr

* Correspondence: qasim.zeeshan@emu.edu.tr (Q.Z.); babak.safaei@emu.edu.tr (B.S.)

**Abstract:** Recently, with the emergence of Industry 4.0 (I4.0), smart systems, machine learning (ML) within artificial intelligence (AI), predictive maintenance (PdM) approaches have been extensively applied in industries for handling the health status of industrial equipment. Due to digital transformation towards I4.0, information techniques, computerized control, and communication networks, it is possible to collect massive amounts of operational and processes conditions data generated form several pieces of equipment and harvest data for making an automated fault detection and diagnosis with the aim to minimize downtime and increase utilization rate of the components and increase their remaining useful lives. PdM is inevitable for sustainable smart manufacturing in I4.0. Machine learning (ML) techniques have emerged as a promising tool in PdM applications for smart manufacturing in I4.0, thus it has increased attraction of authors during recent years. This paper aims to provide a comprehensive review of the recent advancements of ML techniques widely applied to PdM for smart manufacturing in I4.0 by classifying the research according to the ML algorithms, ML category, machinery, and equipment used, device used in data acquisition, classification of data, size and type, and highlight the key contributions of the researchers, and thus offers guidelines and foundation for further research.

**Keywords:** predictive maintenance; artificial intelligence; machine learning; industrial maintenance

---

## 1. Introduction

Industries are currently going through "The Fourth Industrial Revolution," as professionals have called it, a term also known as "Industry 4.0." (I4.0) Integration amongst physical and digital systems of the production contexts is what mainly concerns Industry 4.0 [1]. With the appearance of I4.0, the concept of prognostics and health management (PHM) has become unavoidable tendency in the framework of industrial big data and smart manufacturing; plus, at the same time, it offers a reliable solution for handling the industrial equipment health status. I4.0 and its key technologies play an essential role to make industrial systems autonomous [2,3] and thus make possible the automatized data collection from industrial machines/components. Based on the collected data type machine learning algorithms can be applied for automated fault detection and diagnosis. However, it is very cruel to select appropriate maching learning (ML) techniques, type of data, data size, and equipment to apply ML in industrial systems. Selection of inappropriate predictive maintenance (PdM) technique, dataset, and data size may cause time loss and infeasible maintenance scheduling. Therefore, this study aims to present a comprehensive literature review to discover existing studies and ML applications,

and thus help researchers and practitioners to select appropriate ML techniques, data size, and data type to obtain a feasible ML application.

The industrial equipment predictive maintenance (PdM) can perceive the degradation performance because it was designed to achieve near-zero; hidden dangers, failures, pollution, and near-zero accidents in the entire environment of manufacturing processes [4].

These huge amounts of data collected for ML contains very useful information and valuable knowledge which can improve the whole productivity of manufacturing processes and system dynamics, and can also be applied into decision support in several areas, mainly in condition-based maintenance and health monitoring [5]. Due to the recent advances in technology, information techniques, computerized control, and communication networks, it is now possible to collect vast volumes of operational and processes conditions data generated from several pieces of equipment in order to be harvested in making an automated Fault Detection and Diagnosis (FDD) [6]. The datasets collected can also be applied to develop more efficient methodologies for the intelligent preventive maintenance activities, similarly known as PdM [7].

ML applications provide some advantages which include maintenance cost reduction, repair stop reduction, machine fault reduction, spare-part life increases and inventory reduction, operator safety enhancement, increased production, repair verification, an increase in overall profit, and many more. These advantages also have a tremendous and strong bond with the procedures of maintenance [1,8–11]. Moreover, fault detection is one of the critical components of predictive maintenance; it is very much needed for industries to detect faults at very early stage [12]. Techniques for maintenance policies can be categorized into the following main classifications [13–17].

1.  (R2F) Run 2 Failure: also known as corrective maintenance or unplanned maintenance. It is the simplest amongst maintenance techniques which is performed only when the equipment has failed. It may lead to high equipment downtime and a high risk of secondary faults and thus, create a very large number of defective products in production.

2.  Preventive Maintenance (PvM): also known as scheduled maintenance or time-based maintenance (TBM). PvM refers to periodically performed maintenance based on a planned schedule in order to anticipate the failures. It sometimes leads to unnecessary maintenance which increase the operating costs. The main aim here is to improve the efficiency of the equipment by minimizing the failures in production [18].

3.  Condition-based Maintenance (CBM): this method of maintenance is based on a constant machine or equipment monitoring or their process health that can be carried out only when they are actually necessary. The maintenance actions can only be carried out when the actions on the process are taken after one or more conditions of degradation of the process. CBM usually cannot be planned in advance.

4.  PdM: known as Statistical-based maintenance: maintenance schedules are only taken when needed. It is based on the continuous monitoring of the equipment or the machine, as like CBM. It utilizes prediction tools to measure when such maintenance actions are necessary, hence the maintenance can be scheduled. Furthermore, it allows failure detection at an early stage based on the historical data by utilizing those prediction tools such as machine learning methods, integrity factors (such as visual aspects, coloration different from original, wear), statistical inference approaches, and engineering techniques.

It is required that any maintenance strategy ought to minimize equipment failure rates, must improve equipment condition, should prolong the life of the equipment, and reduce the maintenance costs. An overview for the maintenance classifications is shown in Figure 1. PdM turned out to be one of the most promising strategies amongst other strategies of maintenance that has the ability of achieving those characteristics [19], thus the strategy has been applied recently in many fields of studies. PdM captivates the attention of the industries, hence it has been applied in the era of I4.0 due to it is capability of optimizing the use and management of assets [1,20].

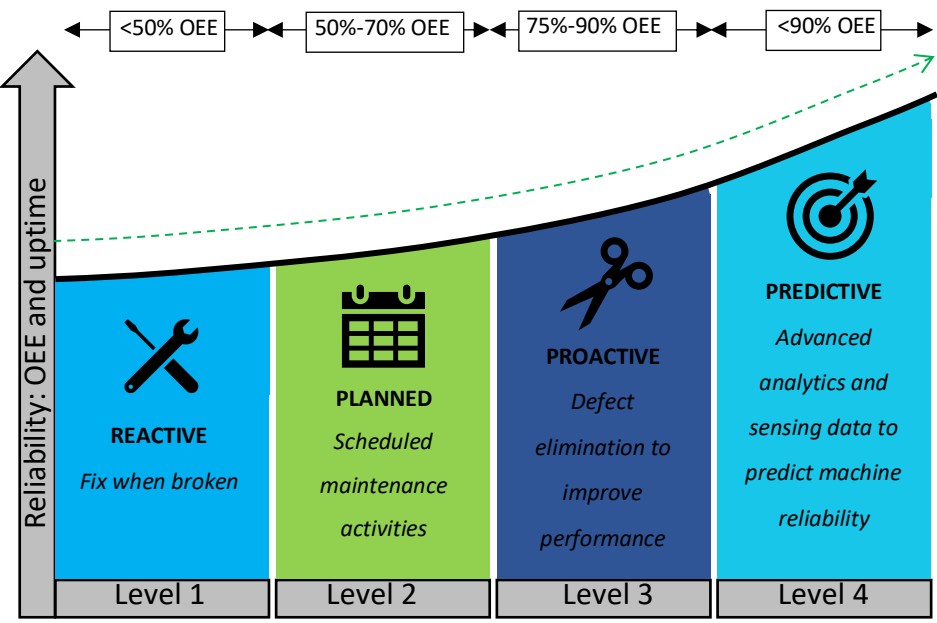

**Figure 1.** Maintenance types [1].

ML, within the contexts of artificial intelligence (AI) (Figure 1, copyright permission of Figure 1 has taken on 20 September 2020), lately, has appeared to be one of the most powerful tools that can be applied in several applications to develop intelligent predictive algorithms. It has been developed into a wide field of research over the past decades. ML can be defined as a technology by which the outcomes can be forecasted based on a model prepared and trained on past or historical input data and its output behavior [21]. According to Samuel, A.L. [22], ML mainly means that if computers are allowed to solve without specifically being programmed in doing so. ML approaches are known to have tremendous advantages, as they have the ability in handling multivariate, high dimensional data and can extract hidden relationships within data in complex, dynamic, and chaotic environments [1,23,24]. However, depending on the ML approach chosen, the performance and advantages might differ. As of today, ML techniques have been widely applied in several areas of manufacturing (such as maintenance, optimization, troubleshooting, and control) [23]. Consequently, this paper aims to provide the recent advancements of ML techniques applied to PdM from an ample perspective. Predominantly, this ample review uses Scopus database while acquiring and identifying the articles used. From a comprehensive perspective, this paper aims to pinpoint and categorize based on the ML technique considered, ML category, equipment used, device used in data acquiring, applied data description, data size, and data type.

The following describes how the paper is organized: firstly, this section gives a brief introduction on the current field of study. Secondly, Section 2 presents a brief background on PdM and ML techniques. Thirdly, Section 3 explains the methodology employed in the literature while considering and categorizing the papers to review and how they are grouped. Section 4 presents the comprehensive applications of ML techniques applied to PdM. Subsequently, discussions are drawn based on the analysis carried out in the literature of ML algorithms for PdM. Finally, a conclusion and future research guidelines are given.

## 2. PdM and ML Techniques

Currently, the PHM system has become a safe-fire method for maintaining the safety status of equipment (e.g., defect detection and Remaining Useful Life (RUL)). It is accomplished by the systematic use of the current testing findings in AI technology and IT technology. [4]. Additionally, PdM cannot only provide reduction in the costs of the maintenance, it can also prolong the RUL [25]. The incipient issues that may lead to disastrous failures can be correctly forecast and appropriate steps

can be set in order to avoid these failures on the basis of the prediction outcomes [4]. Nevertheless, at any appropriate time, an industrial equipment can be replaced or repaired before the fault happens, and thus might restore the original condition of the equipment or system after each and every completed maintenance. Moreover, the equipment, component, system, or a machine's health status can be obtained at any instance. Their failures can also be predicted in order to achieve a non-zero downtime performance [26]. PdM mainly focuses on utilizing predictive info in order to accurately schedule the future maintenance operations [27].

The aim of PdM is not only to collect process data and its parameters, but also to collect the physical health aspects of the equipment, machine, or component (such as pressure, vibration, temperature, viscosity, acoustics, viscosity, flow rate data) and many as such. At the same time, this information collected is now widely used for fault identification, early fault detection, equipment health assessment, and predicting the future state of the equipment [12].

According to [12], ML is a subcategory of AI and is defined as any algorithm or a program that has the ability of learning with the smallest or no additional support. ML assists in solving many difficulties such as in vision, big data, robotics, and speech recognition [12]. Moreover, ML techniques are designed to derive knowledge out of existing data [23,28]. PdM process and technologies to drive PdM is shown in Figure 2. Crucial technologies for PdM such as smart sensors, network, artificial intelligence, big data, and cloud systems are also highlighted by [29,30].

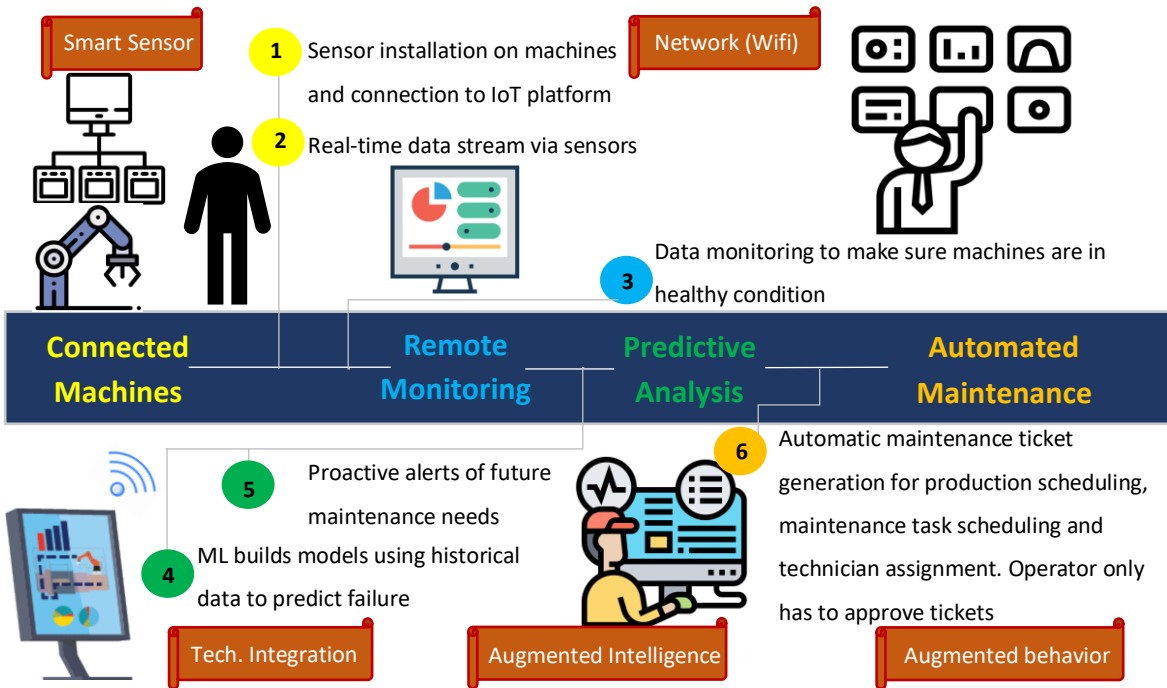

**Figure 2.** PdM process and technologies to drive PdM.

Deloitte [31] classifies the technologies that drive PdM into five different categories. Those are sensors, network, integration, augmented intelligence, and augmented behavior. Smart sensors are used to gather machine information with the use of built-in sensors or environment information with the use of external sensors. The network provides data storage as well as data transfer by using Bluetooth and WiFi [32,33]. Technology integration allows data management and data accumulation via Internet of Things (IoT), augmented intelligence assist data processing, and data analytics [30], whereas augmented behavior allows virtualization, computing and service platform via apps, and tickets to assist the operator [29].

ML algorithms are categorized into three different types; (1) supervised, (2) unsupervised, and (3) reinforcement learning (RL) (see Figure 3) [23,34,35]. The aim is to show how complex the structure

can be and the commonly used available learning techniques. Moreover, as stated by [23], different algorithms can be combined together in order to maximize the classification power. To add on, some among the ML algorithms are both applicable to unsupervised and supervised learning.

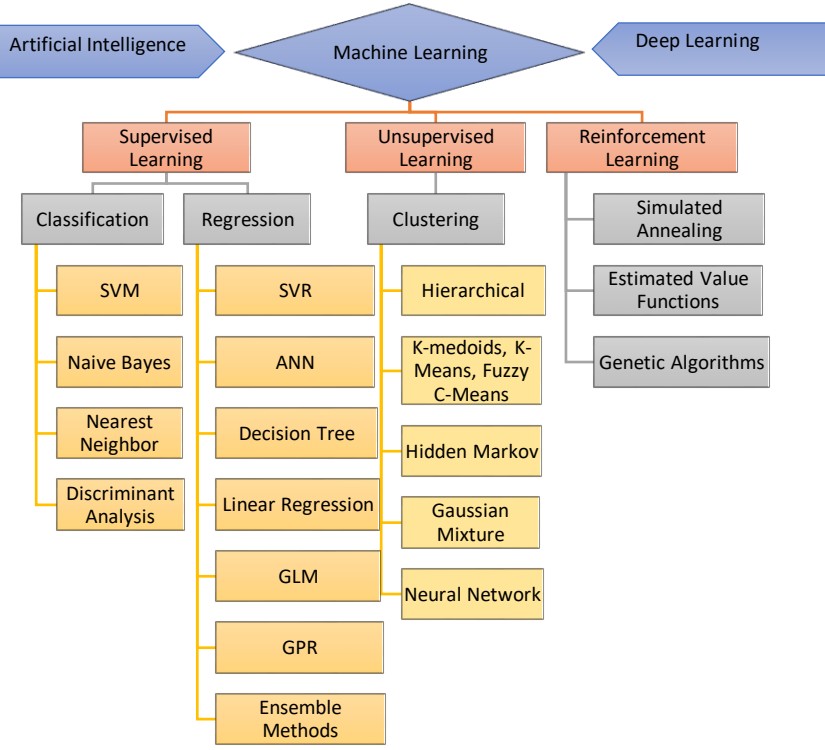

**Figure 3.** Classifications within Machine Learning Techniques.

In unsupervised machine learning, there is no feedback from an external teacher or knowledgeable expert [23]. Based on the existing data, the algorithm identifies the clusters. The main aim here in supervised learning is determining the unknown classes of items by clustering [36], whereas classification is for supervised learning. Unsupervised ML basically defines any ML method that attempts to learn structure in the absence of either an identified output (like supervised ML) or feedback (like Reinforcement learning (RL)) [23].

Clustering, self-organizing maps, and association rules are the basic three main examples of unsupervised learning. In this paper, reviewed articles are categorized into three ML categories, classification, regression, and clustering, as shown in Figure 3. Data utilized in papers are categorized into two data types, real data that are taken from real world machineries, and simulated or synthetic data that are generated to meet specific needs such as model validation in ML. Authors [12,23,32,37,38] has also defined ML categories.

RL is characterized by the delivery of information on training to the community. Through RL, the learner must discover which actions produce the greatest outcomes (numeric reinforcement signal) by attempting instead of being told. [23]. Nonetheless, some of the researchers considered RL as some sort of special supervised learning, like [34]. Moreover, as stated by RL, problems differ from supervised learning, as they can be described by the absence of labeled examples of "good" and "bad" behavior [23].

There are several available supervised machine learning algorithms, as few can be seen from Figure 3. Each of these algorithms has its own specific advantages as well as limitations regarding the application (either PdM or manufacturing). Selecting the most appropriate and suitable ML algorithm can be a major challenge for the requirements of the PdM problem. It is also important to get good at applied machine learning by practicing on lots of different datasets. Therefore, each problem requires different subtlety, different data preparation, and modelling methods. Datasets are classified into

seven categories: multivariate, sequential text, time series, sequential, univariate, text, and domain theory. However, this paper classifies datasets into two categories. One of them is real datasets that are any production data obtained from real production processes and applicable to ML. Another one is synthetic datasets that are any kind of production data applicable to ML, but they are simulated data rather than direct measurement in the production.

Ethical/legal permission is not required for this study. The study complies with research and publication ethics in obtaining all kinds of data and images.

## 3. Survey Methodology and Analysis

The scholarly or academic databases used for this review include articles from Scopus, ScienceDirect, Institute of Electronic and Electrical Engineers (IEEE), and Google Scholar. The Scopus database was mainly used for this review. In this review, the articles reviewed are categorized into two groups. The first group comprises the articles collected from Scopus and are considered as the main featured articles of the research. The second group comprises the articles that are used as supporting or background work in the contexts of introduction and the study in general. The second group of the articles that are obtained from the four databases stated above helped in building the theoretical foundations to the PdM, ML techniques, and the ML algorithms.

Strategy and keywords used while collecting the article from Scopus are as follows:

- Firstly, the search was carried out based on "machine learning."
- Followed by search within search, with "predictive maintenance," note that the use of quotation here means that the whole phrase was searched entirely, not as separate word by word.

With (TITLE-ABS-KEY ("machine learning") AND ("predictive maintenance")), as the search keywords, 788 documents appeared, and the survey was carried out on 30 July 2020.

- The documents were then limited to the recent time parameters of 2010 to 2020. All inclusively, the number of the documents reduced to 367.
- Subject area was limited to engineering, energy, and materials science, then the documents reduced to 273.
- Then, from the document type, review and conference review were excluded from the analysis. That left a total of 217 documents. From the language section, the documents were limited to English.

Figure 4 shows the number of documents that are published over the years, between 2010 and 2020. As can be seen from Figure 4, it is confirmed that just recently (i.e., from the last three years) ML techniques in PdM captivates the attention of the researchers. As there are very few papers published in 2010 and 2011 in comparison to how the number of published articles just spiked-up through the year 2017–2018. Thus, it can be concluded that application of ML technique in the field of PdM is a new method with a growing interest in the field of research. This might be due to the increase in the amount of dataset that are generated in industries, by the industrial equipment, system, or components, and at the same time it could be due to the recent advances of ML techniques and their algorithms [1]. Figure 5 shows the information on how the documents are published by type considered in this review, 134 conference papers, 72 articles, and 11 book chapters.

Among these 217 documents, we were able to download 103. These 103 articles and 33 from IEEE, combined together gives a total of 126 articles. Then, they are analyzed and screened for the removal of any duplicates. The number of publications are given in Figure 4. Article selection criteria aforementioned can be summarized and pictured from the Preferred Reporting Items for Systematic Reviews and Meta-Analyses (PRISMA) flowchart given in Figure 5.

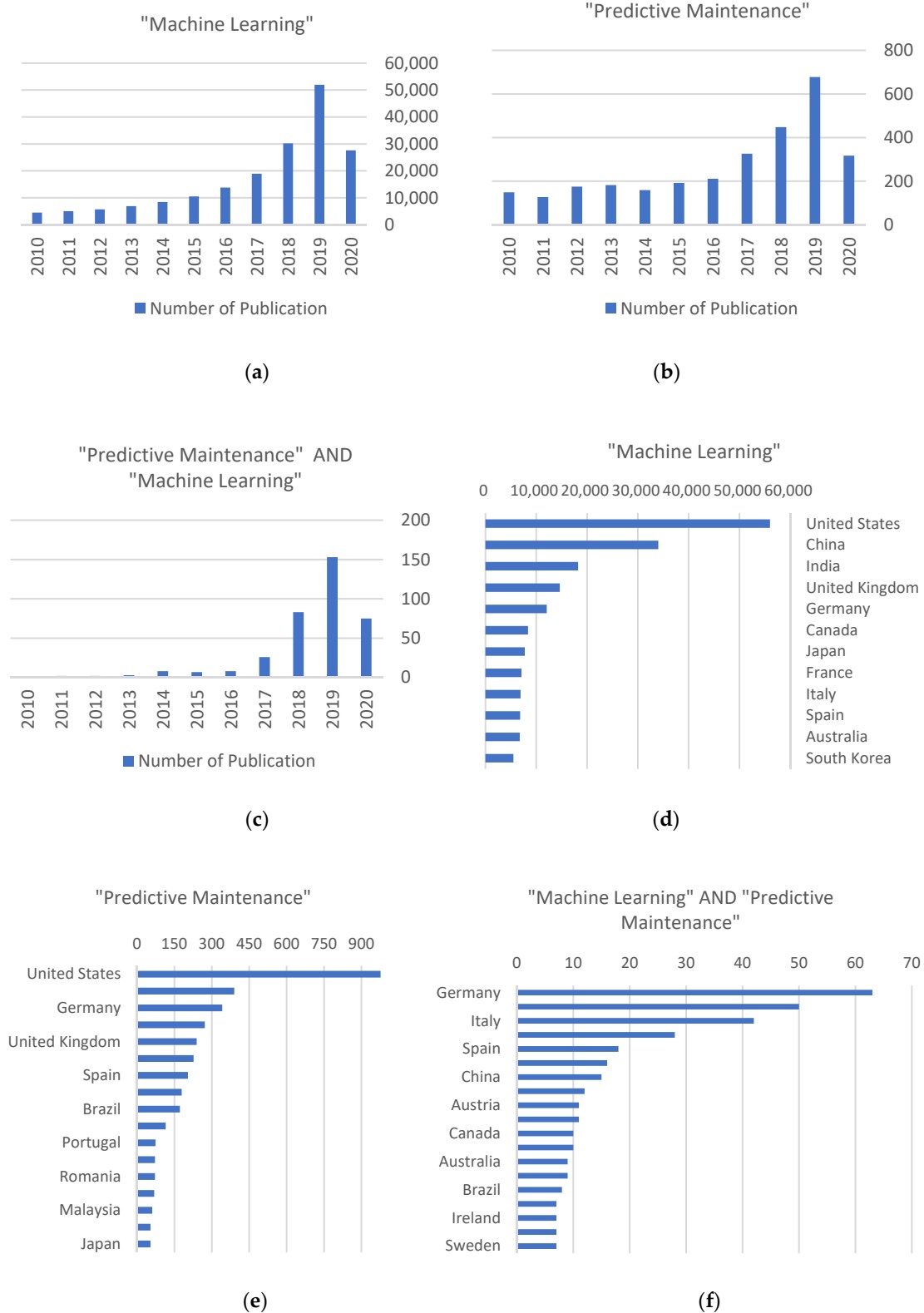

**Figure 4.** Statistics from Scopus database (Date: 30.07.2020), (**a**) Published documents per year with keywords: ("Machine Learning"), (**b**) Published documents per year with keywords: ("Predictive Maintenance"), (**c**) Published documents per year with keywords: ("Machine Learning" AND "Predictive Maintenance"), (**d**) Published documents by country with keywords: ("Machine Learning"), (**e**) Published documents by country with keywords: ("Predictive Maintenance"), (**f**) Published documents by country with keywords: ("Predictive Maintenance" AND "Machine Learning").

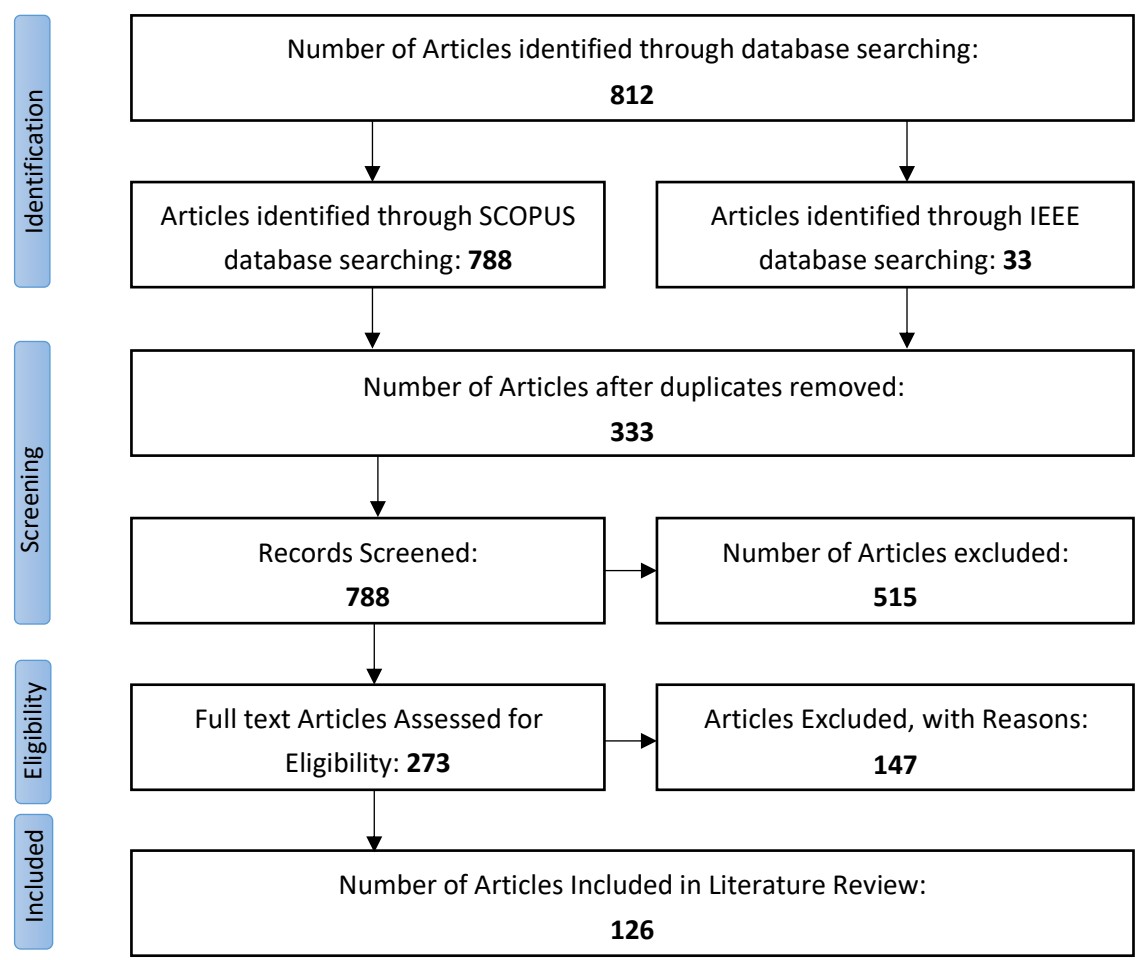

**Figure 5.** Literature survey flowchart.

## 4. Applications of ML Algorithms in PdM

ML algorithms can be used to solve several problems with the enormously available data generated from industries, thus, ML has been widely used in computer science and other areas, such as PdM of manufacturing system, tool or machine, and is one of the possible areas of use for data-driven methods (Artificial Neural Network (ANN), Reinforcement Learning (RF), Support Vector Machine (SVM), Logistics Regression (LR), and Decision Tree (DT)) [4,23]. Recently, ML techniques have been widely applied in various fields of study. Selecting the most appropriate, simple, and the most efficient could be of a great concern. ML algorithms usually require collecting huge amounts of data of the failure status scenarios and the health conditions scenarios for model training. These algorithms that mainly require large amounts of data involves Vector Space Model (VSM), LR, DT, and RF. ML algorithm development covers historical data selection, pre-processing data, model selection, model training, model validation, and maintenance. The steps involved in ML algorithm development can be specified as input, feature extraction & selection, features, traditional ML techniques, and output [4]. Similarly, Ref. [1] describes the main steps for ML development as historical data, data pre-processing, model selection, training and validation, and model maintenance. Further details on the main steps involved can be found in [1]. PdM has been broadly applied in industries such as manufacturing industries using ML techniques [39] and deep learning [40].

### 4.1. Artificial Neural Network (ANN)

In fact, ANN is developed from the subject of biology, where the Neural Network (NN) plays a significant role in the human brain. [41]. ANN is an intelligent computational technique that has

been inspired by biological neurons [10]. It is a massively parallel computing system consisting of an extremely large number of simple processors with many interconnections. Instead of following the set of laws specified by human experts, ANNs learn the basic laws from the set of given symbolic situations in examples [42]. They are organized in three layers or more, (i.e., input layer, several hidden layers, and an output layer) [43]. Moreover, the analytical activity of these ANNs derives from the relations between the network processing units.

ANN models are broadly applied in many fields of studies due to their capability to learn from examples. To add on, ANNs models in comparison to the other traditional machine learning algorithms have noticeable advantages in addressing random data, fuzzy data, and nonlinear data. ANNs are primarily appropriate for systems with a complex, large scale structure and unclear information [4]. ANNs are widely applied and they are the most common ML algorithms [1], at the same time they have been suggested in several industrial applications involving soft sensing [44], and in predictive control systems [45]. Hesser, D.F. and Markert, B. [46] trained an ANN model to classify tool state of a Computer Numerical Control (CNC) milling machine with acceleration data. The proposed study was based on a retrofitting approach in order to facilitate older machines towards to I4.0. The tool wear was monitored by utilizing a programmable prototyping platform equipped with built-in sensors. The study proves the feasibility of retrofitting older machines. In the study, the performance of the built model was compared and outperformed the performance of Support Vector Machine (SVM) and K-Nearest Neighbors (KNN) models.

A methodology was proposed by Sampaio, G.S. et al. [47] to treat and convert the collected data of vibration measurements from a vibration system that simulated a motor and to build a dataset in order to train and test an ANN model capable of predicting the future condition of the equipment, predicting when a failure can happen. The methodology involves the use of frequency and amplitude data by classifying the dataset and defining a way of calculating the vibrating system's failure time. In the study, Multilayer Perceptron (MLP) methodology was used in performing the prediction task, due to its easier implementation with a good generalization index. The ANN model proposed was then compared in terms of its efficiency and based on Root Mean Square Error (RMSE) performance index values with other ML techniques, including Regression Tree (RT), Random Forest (RF), and Support Vector Machine (SVM). Comparative and training results were adequate and showed that ANN was greater than the others. In terms of medium-term and long-term prediction, ANN outperforms the others, whereas generalization in short term predictions between ANN, RF, and RT were equal.

ANN and SVM ML algorithms are applied in developing gauge degradation measurements prediction for two types of rail track including straight and curved segments by Falamarzi, A. et al. [48]. Mean squared error and coefficient of determination are used in the performance evaluation of the proposed models, ANN with greater than 0.9 coefficient of determination value. Based on the results obtained from the study, both ANN and SVM models provide satisfactory and slightly similar outcomes, but the performance of ANN models in predicting gauge deviation of straight segments is slightly better than SVM models. Biswal, S. and Sabareesh, G.R. [10] designed and developed a bench top test-rig for investigating the time domain vibration signatures of several critical components in wind turbine by imitating the operating condition of an actual wind turbine and use it for monitoring its condition. In their work, they acquired the healthy and faulty condition vibration signature of the critical components, then developed and applied ANN model to carry out the classification of the faulty and healthy state features. The model developed shows a 92.6% accuracy classification efficiency. Zhang, Y. et al. [49] reported a study on Physics-based Model and Neural Network Model for Monitoring Starter Degradation of Auxiliary Power Unit (APU). In their study, a generic modeling technique is adopted to overcome the limitations of lack of component characteristics. A comparative analysis between back-spreading and forward-feed neural network models has been performed, trained, and tested. Both models are applied under nominal and deteriorated conditions and their capabilities are validated. Depending on the data collected, their analysis concluded that the physics-based

approach produces more consistent outcomes for cases with degraded starters, although the neural network model showed better results with starters in healthy condition.

### 4.2. Support Vector Machine (SVM)

SVM is a well-known ML technique which is widely used for both classification and regression analysis, due to its high accuracy [1,50,51]. SVM is defined as a statistical learning concept with an adaptive computational learning method. SVM learning algorithm is presented in Figure 6. SVM learning technique employs input vectors to map nonlinearly into a feature space whose dimension is high [52–54].

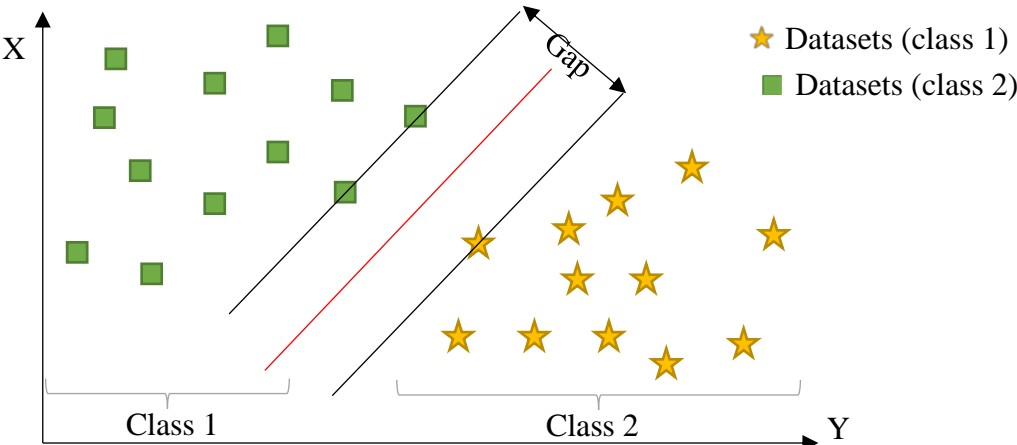

**Figure 6.** Support vector machine algorithm.

SVM is a supervised ML technique that can perform pattern recognition, classification, and regression analysis. In the PdM of industrial equipment, SVMs have been widely applied for identifying a specific status based on the acquired signal [55]. SVM and ANN ML algorithms are applied in developing gauge degradation measurements prediction for two types of rail track including straight and curved segments by Falamarzi, A. et al. [48], where mean squared error and coefficient of determination are used in the performance evaluation of the proposed models, SVM with greater than 0.75 coefficient of determination value. Based on the results obtained from the study, both ANN and SVM models provide satisfactory and slightly similar outcomes, but, the performance of SVM models in predicting gauge deviation of curved segments is slightly better than ANN models. Moreover, in the study, Melbourne tram network has been used as a case study.

A data driven diagnostics and prognostics framework for machines to increase efficiency and reduce maintenance cost was proposed by Xiang, S. et al. [56]. Moreover, an accurate data labeling methodology is developed for supervised learning via comparing the serial number of target components in the adjacent dates. In the study, vending machine real data was used to validate the proposed framework for three different classifiers including SVM, RF, and Gradient Boosting Machines (GBM). Moreover, two models were developed for PdM, one for diagnostics and the other for two-stage prognostics. Results for the cross-validated simulation obtained shows that the diagnostics model can achieve more than 80% of accuracy, thus the developed model of SVM can be applied for diagnosis and prognostics monitoring of complex vending machines. The prognostics model outperforms one-stage conventional prediction models.

### 4.3. Decision Tree (DT)

Decision Tree is a network system composed primarily of nodes and branches, and nodes comprising root nodes and intermediate nodes. The intermediate nodes are used to represent a feature,

and the leaf nodes are used to represent a class label [52]. DT can be used for feature selection [57]. DT algorithm is presented in Figure 7.

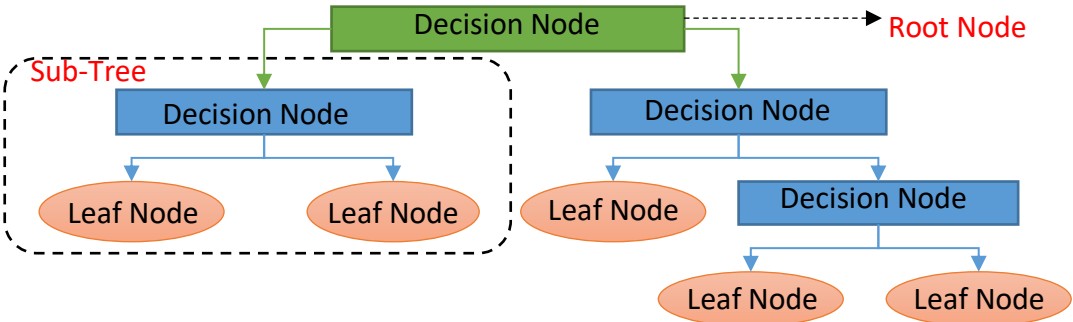

**Figure 7.** Decision tree algorithm, adapted from.

DT classifiers have gained considerable popularity in a number of areas, such as character identification, medical diagnosis, and voice recognition. More notably, the DT model has the potential to decompose a complicated decision-making mechanism into a series of simplified decisions by recursively splitting covariate space into subspaces, thereby offering a solution that is sensitive to interpretation [58,59].

### 4.4. Random Forest (RF)

RF was developed by Breiman, L. [60]. This is an ensemble learning algorithm made up of several DT classifiers, and the output category is determined collectively by these individual trees. When the number of trees in the forest increases, the fallacy in generalization error for forests converges. There are also important benefits of the RF. For example, it can manage high-dimensional data without choosing a feature; trees are independent of each other during the training process, and implementation is fairly simple; however, the training speed is generally fast and, at the same time, the generalization functionality is good enough [4]. Random forest algorithm for machine learning has tree predictions, and based on tree predictions, the RF provides random forest predictions [61]. The RF model is visualized in Figure 8.

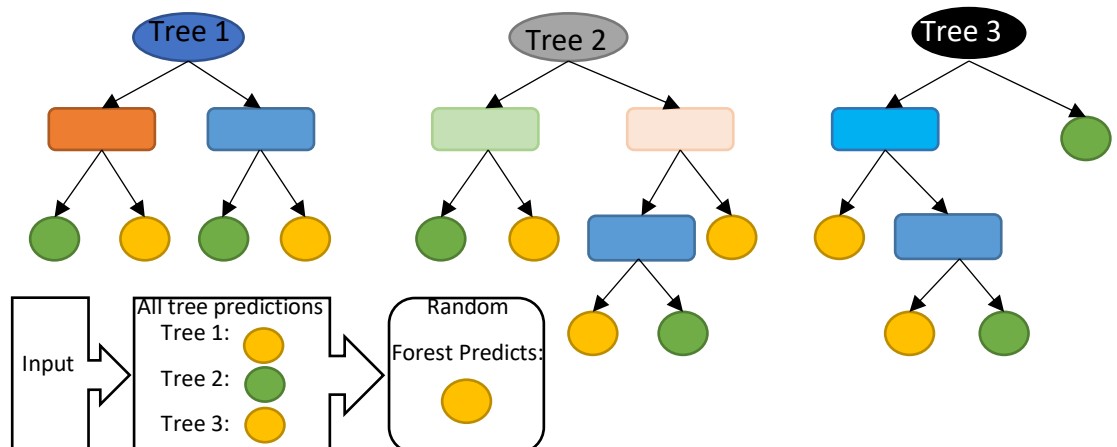

**Figure 8.** Random forest regression algorithm for machine learning.

A study was reported on forecasting the downtime of a printing machine based on real time predictions of imminent failures [62]. In their study, they used unstructured historical machine data to train the ML classification algorithms including RF, XGBoost, and LR to predict the machine failures. Different metrics were analyzed to determine the fitness of the models. These metrics

include empirical cross-entropy, area under the receiver operating characteristic curve (AUC), receiver operating characteristic curve itself (ROC), precision-recall curve (PRC), number of false positives (FP), true positives (TP), false negatives (FN) and true negatives (TN) at various decision thresholds, and calibration curves of the estimated probabilities. Based on the results obtained, in terms of ROC, all the algorithms performed significantly better and almost similar. But in terms of decision thresholds, RF and XGBoost perform better than LR.

ML algorithms including Linear Regression, RF, and Symbolic Regression (SR) are applied in modeling the condition of a healthy industrial machinery [63]. They proposed a methodology for detecting and predicting drifting behavior (called concept drifts) in continuous data streams. Further, a real-world case study was presented on industrial radial fans. Based on the results obtained using the synthetic data, both the results from concept drift detection and prediction are highly successful. Moreover, based on the conducted real word study, experts on-site at the strained radial fan reported that the principle of drift detection has been successfully deployed. However, due to the lack of continuous deterioration information, the predictability of concept drifts is currently based on assumptions and cannot be measured yet, even though results of the tests are already very promising.

Janssens, O. et al. [64] proposed a multi-sensor device that uses not only infrared thermal imaging data, but also uses vibration measurements for automatic conditioning and fault detection in rotating machines. The feature fusion is utilized where model-driven features are derived from vibration measurements and data-driven features are derived from infrared thermal imaging data. Then, the extracted features are combined together and presented to RF classifiers for actual fault detection. They have demonstrated in the study by mixing these two types of sensor data, a variety of conditions/faults and combinations can be measured more accurately than in the case of individual sensor streams.

Lacaille, J. and Rabenoro, T. [65] developed a learning algorithm that can automatically detect and analyze multidimensional datasets of turbofan engine. The developed model uses a very wide population of pre-treatments and statistic tests on the data and has the ability to select good combinations of tests with higher than 85% pre-identification. Quiroz, J.C. et al. [66] proposed a new approach to diagnose broken rotor bar failure in a Line Start-Permanent Magnet Synchronous Motor (LS-PMSM) using RF. The transient current signal during the motor startup was acquired from a healthy motor and a faulty motor with a broken rotor bar fault. The model was trained using features extracted from thirteen different statistical time-domain features, and these features were used in determining the state of the motor where it is operating under faulty or normal conditions. Feature importance was considered for their feature selection in order to reduce the number of features to very few from the RF. Results have shown that RF categorizes the motor disorder as safe or deficient with an accuracy of 98.8% using all the features and an accuracy of 98.4% using only the mean index and impulsion features. A comparison was carried out between the developed model and other traditional ML algorithms including Decision Tree (DT), Naive Bayes classifier (NBC), LR, linear ridge, and SVM, the RF consistently outperforms these algorithms with having a higher accuracy than the other algorithms. The suggested methodology can be used for electronic tracking and fault detection of LS-PMSM motors in the industry, and the findings can be beneficial for the development of preventive maintenance plans in factories.

Yan, W. and Zhou, J.H. [67] proposed a predictive model using Term Frequency-Inverse Document Frequency (TF-IDF) and RF can forecast faults of high sensitivity in advance by analyzing the historical data of aircraft maintenance systems, and preventive maintenance may be carried out on the basis of the model's prediction performance. TF-IDF has been employed in order to extract the features from raw data in the past consecutive flights. Different priorities were considered in classifying the faults by the proposed RF model. The ROC curve has been adopted as a performance metric as the dataset is highly imbalanced. Compared to the other method, the suggested approach reaches the maximum true positive rating of 100% and the lowest false positive rate of 0.13%. For the testing dataset, the proposed method achieves true positive rate 66.67% and false positive rate 0.13%.

### 4.5. Logistic Regression (LR)

Binding, A. et al. [62] reported a study on forecasting the downtime of a printing machine based on real time predictions of imminent failures. In their study, they utilized unstructured historical machine data to train the ML classification algorithms including RF, XGBoost, and LR in predicting the machine failures. Various metrics were analyzed to determine the goodness of fit of the models. These metrics include empirical cross-entropy, area under the receiver operating characteristic curve (AUC), receiver operating characteristic curve itself (ROC), precision-recall curve (PRC), number of false positives (FP), true positives (TP), false negatives (FN), and true negatives (TN) at various decision thresholds, and calibration curves of the estimated probabilities. Based on the results obtained, in terms of ROC, all the algorithms performed significantly better and almost similar. But in terms of decision thresholds, RF and XGBoost perform better than LR. Using a given set of independent variables, linear regression is used to estimate the continuous dependent variations. However, using a given set of independent variables, logistic regression is used to estimate the categorical contingent variations [68]. Graph of the linear regression model and logistics regression model are shown in Figure 9.

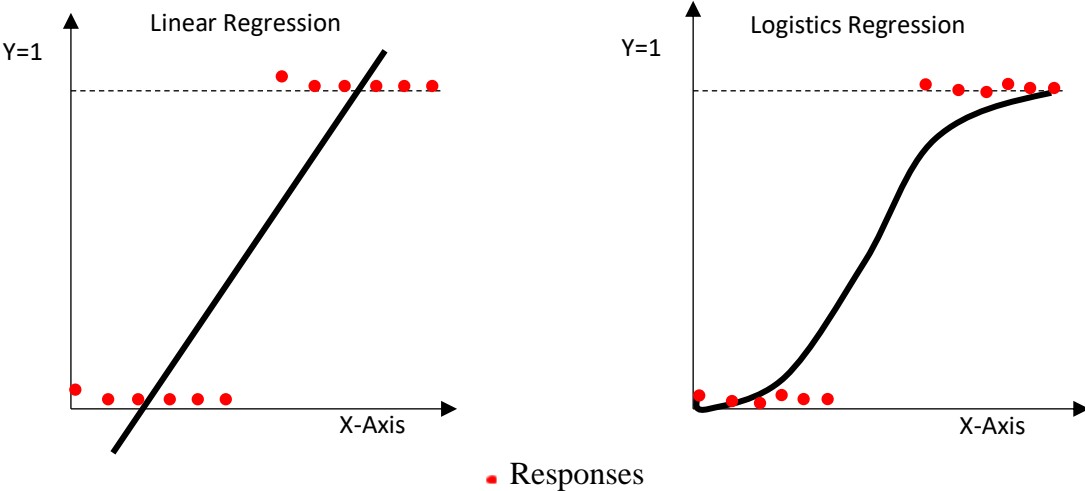

**Figure 9.** Logistic regression.

### 4.6. Extreme Gradient Boosted Trees (XGBoost)

XGBoost was developed by Chen, T. & Guestrin, C. [69], a scalable tree boosting system that is widely used by data scientists and provides state-of-the-art results on many problems. Open source C++ was utilized in the implementation of XGBoost algorithm on forecasting the downtime of a printing machine based on real time predictions of imminent failures [62] and used unstructured historical machine data to train the ML classification algorithms including RF, XGBoost, and LR in predicting the machine failures. Various metrics were analyzed to determine the goodness of fit of the models. These metrics include; empirical cross-entropy, area under the receiver operating characteristic curve (AUC), receiver operating characteristic curve itself (ROC), precision-recall curve (PRC), number of false positives (FP), true positives (TP), false negatives (FN) and true negatives (TN) at various decision thresholds, and calibration curves of the estimated probabilities. Based on the results obtained, in terms of ROC all the algorithms performed significantly better and almost similar. But in terms of decision thresholds, XGBoost and RF perform better than LR. XGBoost algorithm tree uses majority voting technique to define final class [70]. XGBoost algorithm tree is presented in Figure 10.

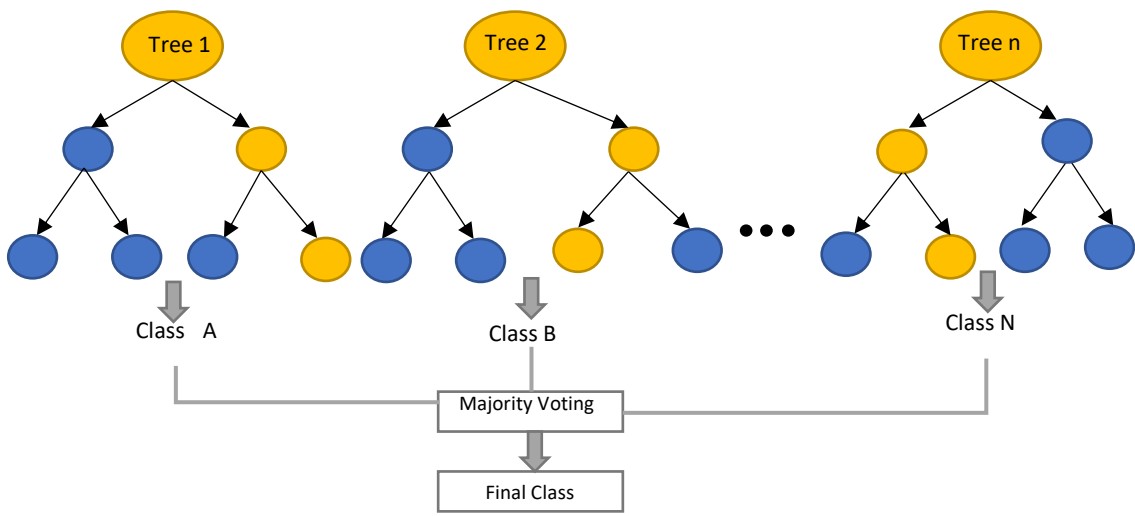

**Figure 10.** XGBoost algorithm tree.

### 4.7. Gradient Boosting Machines (GBM)

GBM is a family of powerful machine-learning techniques that have shown considerable success in a wide range of practical applications. It is also an assembly-based model that learns to update prediction results on new models consecutively [71,72].

A data driven diagnostics and prognostics framework for machines to increase efficiency and reduce maintenance cost was proposed by Xiang, S. et al. [56]. Moreover, an accurate data labeling methodology is developed for supervised learning via comparing the serial number of target components in the adjacent dates. In the study, vending machine real data was used to validate the proposed framework for three different classifiers including SVM, RF, and Gradient Boosting Machines (GBM). Moreover, two models were developed for PdM, one for diagnostics and the other for two-stage prognostics. Results for the cross-validated simulation obtained shows that the diagnostics model can achieve more than 80% of accuracy, thus the developed model of GBM can be applied for diagnosis and prognostics monitoring of complex vending machines. The prognostics model outperforms one-stage conventional prediction models.

### 4.8. Linear Regression

Linear regression refers to a multivariate linear combination of regression coefficients [63]. The coefficients are calculated by the generalized least square technique. Linear regression is deterministic and the parameter is less, there is no need to adjust something other than the data break for model training and testing. Linear regression and Random Forest Regression are very common regression algorithms in general and time series regression algorithms that have already been used in the field of predictive maintenance [73]. Linear regression model in machine learning is presented in Figure 11.

ML algorithms including linear regression, RF, and Symbolic Regression (SR) are applied in modeling the condition of a healthy industrial machinery [63], where they proposed a methodology for detecting and predicting drifting behavior (called concept drifts) in continuous data streams. Further, a real-world case study was presented on industrial radial fans. Based on the results obtained using the synthetic data, both the results from concept drift detection and prediction are highly successful.

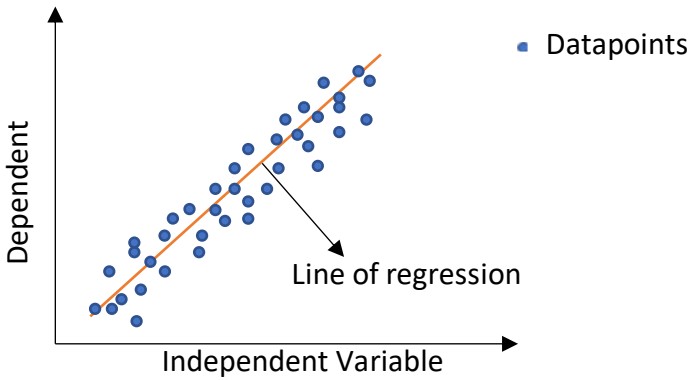

**Figure 11.** Linear regression in machine learning.

### 4.9. Symbolic Regression (SR)

SR refers to models in the form of a syntax tree composed of arbitrary mathematical symbols (terminals: constants and variables, non-terminals: mathematical functions) that can be easily converted into simple mathematical functions. Top-down syntax trees are reviewed for target estimation [63]. Syntax trees are developed using the stochastic genetic programming technique from the field of evolutionary algorithms [74].

SR has been applied in modeling the condition of a healthy industrial machinery [63]. The study proposed concept drifts methodology in continuous data streams. Further, a real-world case study was presented on industrial radial fans. Based on the results obtained using the synthetic data, both the results from concept drift detection and prediction were highly successful. Sample of SR algorithm is shown in Figure 12.

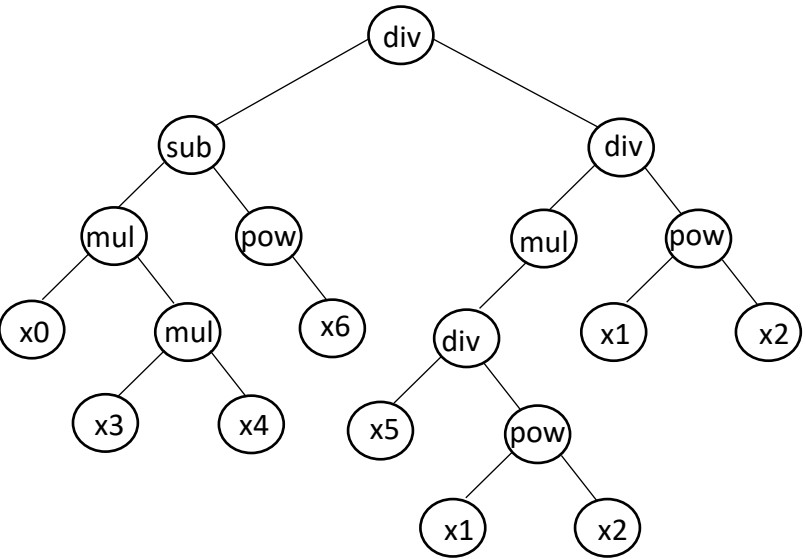

**Figure 12.** Example of SR algorithm.

### 4.10. Other Machine Learning Algorithms

Janssens, O. et al. [75] reported a study on DL for Infrared Thermal Image Based Machine Health Monitoring, where in the study they considered Convolutional Neural Network (CNN), a Feature Learning (FL) tool, in detecting the various conditions of the machine. Moreover, FL was considered because it requires no feature extraction nor expert knowledge. Transfer Learning (TL) is a means to reuse layers of a pretrained Deep Neural Network (DNN). This was mitigated in their study. Case studies have been carried out on machine-fault detection and the oil-level prediction; in both

cases, results have shown that CNN outperforms the classical FE methods. They added that the proposed method has the potential to improve online CM, like offshore wind turbines. Another potential application is the monitoring of bearings in manufacturing lines. Using thermal imaging together to the trained CNN allows identifying the location of the faults in the manufacturing lines.

Huuhtanen, T. and Jung, A. [76] proposed a study on DL for predictive maintenance of photovoltaic panels. CNN was applied for monitoring the operation of photovoltaic panels. In fact, they estimate the regular electrical power curve of the photovoltaic panel depending on the power curves of the neighboring panels. An unusually broad difference between the predicted and the actual (observed) power curve can be used to suggest a malfunctioning panel. By the means of numerical experiments, they are able to demonstrate that the proposed method is able to predict accurately the power curve of a functioning panel and the method out-performs the existing methods that are based on simple interpolation filters.

Pan, Z. et al. [77] proposed a modular cognitive acoustics analytics service for IoT that provides customers with an incremental learning approach to improve their analytical capabilities for non-intuitive and unstructured acoustic data through a combination of acoustic signal processing. They pointed out that different types of data formats created from complicated acoustic environments can go through pre-processing and noise reduction stages and then feed into higher-level analytics platforms. The model allows for acoustic signal-based anomaly detection, acoustic grouping, acoustic signal processing, acoustic array processing, and other features. In classification, the model uses a baseline algorithm when a small amount of data is used, while when a huge amount of data is used, this model utilizes a technique based on the DNN to perform a more accurate classification. This service will include signal processing data, such as sound intensity, spectral centroid, frequency, etc., and can support numerous applications. Eventually, the service can also detect several sources of sound that allows detection and enhancement of the acoustic source. Experimental findings show that this service achieve excellent performance. The application case for the diagnosis of a washing machine is defined.

Jimenez-Cortad, A. et al. [78] carried out a case study based on the application of predictive maintenance to a real machining process. The aim of their study is to increase tool life of the machine by application of ML methods for RUL prediction. Real-time data obtained from the computer and then approximation of the data performed in the analysis. Their study conducted linear and quadratic regression models to perform the design application for RUL estimation. Finally, accurate results were found in their study to predict RUL for comparison. Luo, W. et al. [79] employed predictive maintenance approach for machine tool driven by digital twin to avoid faults and causalities. In their study, a hybrid approach was utilized to calculate RUL results that show the prediction error ratio (between actual value and predicted value).

In this section, a summary on the applications of ML algorithms in PdM will be given. Table 1 summarizes the analysis and comparison among these algorithms that are mainly applied in the field of PdM according to ML techniques, ML categories, equipment systems, and type of data.

**Table 1.** Recent Applications of machine learning (ML) in Predictive Maintenance.

| ML Techniques | ML Cat. | Equipment/ System | Device Used for Data Acquisition | PdM Data Description | Data Size | Data Type | Key-Findings | Ref |
|---|---|---|---|---|---|---|---|---|
| ANN | C | Tool wear for CNC-MM, Deckel Maho DMU 35M | Bosch XDK sensor | Acceleration data | 3-dimensional input vector | Real data | • Tool wear monitoring of a CNC-MM with equipped built-in sensors. • Applicable to older machines that can be utilized in I4.0. • Explore and enable a rapid adaptation to new environmental conditions. • It can be used to predict RUL of the tool. | [46] |
| ANN | C | AK-FN059 with 12 cm cooilng fan | MMA8452Q-Accelerometer | Motor vibration measurements | 9180 observations with 4 attributes | Synthetic data | • Generates a training dataset based on vibration measurements. • ANN trained to predict equipment failure time. • k-fold cross-validation and model generalization performed. • Compared with other ML techniques. Resulted that ANN performs better. • In comparison to RT, RF, and SVM, ANN shows better results. | [47] |
| LR XGBoost RF | C | Printing machine | - | Machine's operational status data | 100 operational variables/minute | Real data | • The fit of the models was determined by various metrics. • Based on decision thresholds, RF and XGBoost perform better than LR. • All the algorithms performed similarly better in terms of ROC. | [62] |
| ANN SVM | C, R | Rail-Tram track, 250 km of double tracks and 25 routes | Non-contact optical laser | Track geometry data-gauge measurements data | - | Real data | • Used for tracking gauge deviation and measurements prediction. • Slightly ANN models perform better in predicting the gauge deviation of straight segments. • SVM models are better in predicting gauge deviation of curved segments. | [48] |

**Table 1.** *Cont.*

| ML Techniques | ML Cat. | Equipment/ System | Device Used for Data Acquisition | PdM Data Description | Data Size | Data Type | Key-Findings | Ref |
|---|---|---|---|---|---|---|---|---|
| ANN | C | Wind turbine at 1200 rpm | Accelerometer sensor | Vibration signals data | 243 datasets with 10,000 sample length | Real data | <ul><li>Investigated the time domain vibration signatures for critical components.</li><li>Acquired the healthy and faulty condition vibration signature.</li><li>Model classifies faulty and healthy state features with 92.6% classification efficiency.</li></ul> | [10] |
| DNN-CNN, DL | C | Rotating machinery | IRT image, temperature sensors | Accelerometer, thermocouple, and thermal camera measurements | 5 REBs × 8 conditions for IRT and 5 REBs × 12 temperature measurements | Real data | <ul><li>CNN algorithm applied to detect several conditions of rotary machinery.</li><li>The technique is able to improve online CM in offshore wind turbines.</li><li>Can be applied in manufacturing lines to monitor bearings.</li><li>Oil-level prediction and machine-fault detection CNN outperforms.</li><li>FL technique provides 6.67% better compared to the FE technique.</li></ul> | [75] |
| CNN | C | Photovoltaic panels | Wireless sensor | Electrical power signals | 400 samples at 1-min sample interval for 1000 days | Synthetic data and real data | <ul><li>CNN techniques can be applied in PdM of PV panels.</li><li>Validated with the means of numerical experiments, that can accurately predict the power-curve of functioning panel.</li></ul> | [76] |
| CNN | C | IoT- IBM system x3650 M4 machine that has 16 hardware threads with 192 GB memory | Acoustic sensor | Acoustic sensor data | Dataset of >20 h-audio within 7-sounds of classes | Real data | <ul><li>Model allows acoustic classification, signal based anomaly detection, signal processing, sensor array processing.</li><li>Uses deep-NN to provide more precise classification when large amount of data is used.</li><li>Presented excellent performance and it can provide signal analysis results.</li></ul> | [77] |

**Table 1.** *Cont.*

| ML Techniques | ML Cat. | Equipment/ System | Device Used for Data Acquisition | PdM Data Description | Data Size | Data Type | Key-Findings | Ref |
|---|---|---|---|---|---|---|---|---|
| NN | - | APU-gas turbine, GTCP 331–430 kW | Fleet of Airbus | Gas path measurements | 310 routine records, with 14-starter nominal and 12-starter degraded conditions | Real data | • Developed an NN and physics-based models for starter degradation monitoring.<br>• Both models are able to monitor the health of the starter and can illustrate symptoms of degradation.<br>• NN provides more accurate results with the starters at healthy condition.<br>• Physics-based provides better results on the starters with growing degradations. | [49] |
| BN | C | TASMI-01 | - | Semiconductor manufacturer | 1890-time intervals with 24 failure occurrences | Real data | • Failure predictions performed by BN for event-driven maintenance data.<br>• The promising results obtained from offline prediction of the case study reveals the significance to outspread it for real time predictions.<br>• The BN model can be used for fault diagnosis not only for failure inference. | [80] |
| RF | C | Rotating machinery with SEW Euro drive 1.1 kW motor | IRT imaging and Vibration-based sensors | Thermal imaging and vibration data | 5 bearings × 8 conditions for IRT and 5 bearings × 12 measurements | Real data | • A multi-sensor system for rotating machinery using a feature fusion method.<br>• Multi-sensor technique can compensate the shortcomings of the heat or vibrations.<br>• Offers a significant increase in fault detection performance. | [64] |
| RF, DT, NB, BC | C | Turbofan-engine | Phonic wheel-pressure, fuel metering valve, temperature sensors | Turbofan-engine multidimensional data | 1616 observations of abnormal and healthy engines | Real data | • Developed a learning algorithm that automatically detects and analyzes multidimensional dataset of turbofan-engine.<br>• The model has the ability to select good combinations of tests with higher than 85% pre-identification. | [65] |

**Table 1.** *Cont.*

| ML Techniques | ML Cat. | Equipment/ System | Device Used for Data Acquisition | PdM Data Description | Data Size | Data Type | Key-Findings | Ref |
|---|---|---|---|---|---|---|---|---|
| RF, -DT, NBC, SVM, LR, linear ridge | C | Rotor bar-LS-PMSM | Torque and speed sensors | Transient current signal data | 320 tests for healthy and faulty motor | Real data | • Uses RF to detect a broken rotor bar failure in LS-PMSM. <br> • RF classifiers results outperformed compared to other techniques result. <br> • Validity and reliability of the RF technique for fault detection performed. <br> • The technique attained 98.8% correct rate of diagnosis. <br> • Impulsion and mean-index are identified. | [66] |
| RF, TF-IDF -SVM, LR | C | Aircrafts | - | Historical data of aircraft maintenance systems | 1542 events with 131 fault types for 2 years | Real data | • ML model for fault prediction using TF-IDF and RF of aircraft. <br> • Used TF-IDF to extract the features of the raw-dataset from past flights. <br> • RF achieves 66.67% true positive rate with a low of 0.13%. | [67] |
| RF, and XGBoost | C | Production lines and semiconductor | - | Bosch and SECOM manufacturing datasets | Bosch: 1.2 mill observations and over 4000 features. | Real data | • Analytics framework applied in fault detection to Bosch and SECOM datasets. | [81] |
| GLM, RF, GBM, DL | - | Semiconductor | Tool sensors, process recipe and wafer count. | Past maintenance, tool sensors, and process datasets | 21 tool-sensors, 3 process recipe and wafer count | Real data | • Ensemble learning selected as best fit technique for PdM for semiconductors. <br> • PdM strategies on ML models and equipment data. <br> • Emphasized the need to adopt cross validation on ensemble PdM based models. | [82] |
| SAFE | R | Semiconductor Ion Implantation process | 31 sensors, - ion-implantation tool | Maintenance cycles dataset | 33 R2F maintenance cycles with 31 variables | Real data | • An approach that dealt with time-series data extraction for PdM. <br> • Applied SAFE methodology in time-series PdM. <br> • The technique outperforms the classical feature extraction methods. | [83] |

**Table 1.** *Cont.*

| ML Techniques | ML Cat. | Equipment/ System | Device Used for Data Acquisition | PdM Data Description | Data Size | Data Type | Key-Findings | Ref |
|---|---|---|---|---|---|---|---|---|
| RF | C | Industrial pumps | Accelerometer and temperature sensors - ABB WiMon 100 | Industrial pumps vibration data | 30 chemical plant industrial pumps for 2.5 years with 1066 features. | Real data | • A case study based on real data of industrial plant. <br> • RF algorithm that can detect faults in vibration data for up to 7 days. <br> • The model was validated using data collected for over 2.5 years. | [84] |
| RF | C | Industry 4.0 - cutting machine | Machine PLCs and communication protocols sensors | PLCs and communication protocols sensors data | 530,731 data readings for 15 different machine features | Real data | • Permits dynamical decision rules for PdM. <br> • Shows a suitable performance in predicting the machine stages with good precision. <br> • Predicts different stages of machine with 95% of high accuracy. | [85] |
| RF | C | Refrigeration systems of supermarket | Temperature sensors | Temperature, work-order data and defrost state | 2265 refrigeration cases across 17 stores for 2 months | Real data | • ML based model for early detection in refrigeration system. <br> • Validated with a real data for 2265 different refrigerators. <br> • Achieved 89% in precision. <br> • A lead time of 7 days. <br> • A recall of 46% when evaluated on unseen cases. | [86] |
| RF | C | HDD | - | Historical data | Dataset of 16,862 failures and 47,677 non-failures for 2 years | Real data | • PdM of HDD failure detection based on Apache Spark. <br> • The model can assist IT staff by making them more proactive and productive by identifying imminent disk failure quickly. | [87] |
| RF | C | Induction motor, 2.2 kW | Datalogger with WIFI communication capability | Voltage and Current waveforms data | 3-phases dataset of 1159-(357-healthy and 802-faulty) | Real data | • Analyzed single and double classifier approach. <br> • Methods could effectively be used in detecting the inter-turn short- circuits using few numbers of data points. <br> • Double classifier approach produces a better result than single. | [88] |

**Table 1.** *Cont.*

| ML Techniques | ML Cat. | Equipment/ System | Device Used for Data Acquisition | PdM Data Description | Data Size | Data Type | Key-Findings | Ref |
|---|---|---|---|---|---|---|---|---|
| RF | C | Wind turbine | Alarms activations and deactivations | Alarm types and operational dataset for 17-turbines | 448-alarm types and 104-operational data (2 years-1,787,040 dataset) | Real data | • Approach to generate a predictive data driven models based upon historical dataset.<br>• A front-end where the status of the turbine can be visualized in real time.<br>• Experiments proved that the implementation has gained an optimal overall success. | [89] |
| RF | C | Trucks and buses air compressors | Logged on-board | LVD and VSR dataset with 65,000 European Volvo trucks | 1250 unique features | Real data | • Commercial vehicles air compressor to detect forthcoming faults.<br>• Generalizes to repair several components of a vehicle.<br>• Used RF and two techniques for feature selection.<br>• The ML feature-based model outperform compared to the human ones.<br>• Usage sets and Beam search 1 were found to be the best features.<br>• A positive profit was shown by all the features at final evaluation. | [90] |
| LSTM | Classifier | Turbofan engine | C-MAPSS tool – multiple sensors | NASA Ames Prognostics Data Repository | A dataset of 4subsets with 708-trajectories by 21 columns for 21 sensors | Synthetic data | • A DPM framework can be used for pro-prognostics decisions and prognostics estimations.<br>• No specific degradation model or a specific RUL function.<br>• Does not predict the RUL.<br>• It provides the probabilities of when the system will fail into several time intervals.<br>• Model that permits maintenance costs and inventory evaluation. | [91] |

**Table 1.** *Cont.*

| ML Techniques | ML Cat. | Equipment/ System | Device Used for Data Acquisition | PdM Data Description | Data Size | Data Type | Key-Findings | Ref |
|---|---|---|---|---|---|---|---|---|
| LSTM | Clustering | Engine | Operational and Sensors | CMAPSS NASA simulation dataset of engine degradation | 14-inputs and 4-outputs of (21 sensor measurements and 3-operational settings) | Synthetic data | • Implemented an LSTM model to Apache Spark on a large-scale dataset for current life condition predictions of an engine. • Deals with sequential data. • LSTM network output is to decide the current state of a component before their life ends. • Trained and tested on an engine degradation open source data. • It can used in industries to detect break downs before they occur. | [92] |
| 1NNC, LS-SVM-linear kernal | C | Rolling bearing | TFI recognition | Vibration signals data | 80 samples for a test with 1024 data points. | Real data | • Developed an approach for fault monitoring in bearings based on sparse TFIs recognition. • Traditional time–frequency drawbacks can be overcome by applying STFA-PD. • STFA-PD method shows a promising diagnosing performance. • Used to obtain high-quality TFIs and also used in analyzing non-stationary signals. | [93] |
| SVM, - PF, GAPF | R | Aircraft | LVDT and RVDT | Actuator internal oil leakage fault data | 2400- training samples and 480- testing samples datasets | Real data | • Studied a novel oil leakage fault prognostics method for actuators. • Presented a hybrid SVM-PF framework for actuator fault prognostics. • Tested and the results obtained as a satisfactory performance. • SVM-PF hybrid framework has better prognostics accuracy. • Higher fault resolution was observed compared to traditional ML. | [94] |

**Table 1.** *Cont.*

| ML Techniques | ML Cat. | Equipment/ System | Device Used for Data Acquisition | PdM Data Description | Data Size | Data Type | Key-Findings | Ref |
|---|---|---|---|---|---|---|---|---|
| SVM, RF, GBM | R | Vending machines | - | System logs, operation logs, context, process and performance logs data. | 3450 malfunction machines data | Real data | • Framework for machines to increase efficiency and reduce maintenance cost.<br>• Used vending machines real data to validate the proposed framework.<br>• Two models developed for PdM, one for diagnostics and the other for two-stage prognostics.<br>• The diagnostics model can achieve more than 80% of accuracy.<br>• The prognostics model outperforms one-stage conventional prediction models. | [56] |
| SVM, Binary LR, Gamma process | C, R, D | Track | - | Track geometry data of railroads | 6500 red and 17,500 yellow tag defects records of dataset | Real data | • Suggested that ensemble methodology outperform the other techniques.<br>• A technique for forecasting track geometry defects by developing an ensemble classifier.<br>• In all cases, ensemble classifier proved to outperform the individual algorithms.<br>• The technique can be applied to any type of tracks and other type of defects. | [95] |
| SVM, RF, LDA | C | Railways -Mile track | - | Track geometry data | 1 mile of track with 28 inspection dates and 31 features by 31 parameters | Real data | • Investigated the possibility of minimizing multivariate track geometry indices into a low-dimensional form.<br>• SVM was found to be the most effective technique and predicts better track defects.<br>• Measured the performance of the model using TPR and FPR. | [96] |

**Table 1.** *Cont.*

| ML Techniques | ML Cat. | Equipment/ System | Device Used for Data Acquisition | PdM Data Description | Data Size | Data Type | Key-Findings | Ref |
|---|---|---|---|---|---|---|---|---|
| SVM-Regression kernel | R | Gas turbine engine of an aircraft | Time series sensor | Sensor measurements data of the time series from CMAPSS dataset of aircrafts | CMAPSS dataset-14 inputs with 21 outputs | Synthetic data | • Model for SVM in prognostics prediction of multiple time series tasks. <br> • Testing was carried out with simplified time-series simulated data. <br> • Improvements were shown from the results obtained over the conventional SVM results. | [97] |
| 1st and 2nd class SVM, DT, RF, LSTM-NN | - | Power Transformer | - | Insulating oil tests dataset: Mechanical and chemical measurements | 15,031 tests with 30 variables | Real data | • Used insulating oil tests dataset for failure prediction in power transformers using several ML techniques. <br> • SVM produces best performance among the other tested techniques. <br> • SVM achieved 77% recall performance with 35% FPR. <br> • LSTM did not give better performance because the data was collected at low frequency. <br> • Collecting the data at an adequate frequency is better than large number of observations. | [98] |
| SVM, K-means, K-NN, Euclidean Distance, and CRA | C | Wind turbine bearings | Accelerometer, displacement, velocity, and torque sensors | Vibration signals data | - | Real data | • Applied several ML methods for bearing fault detection. <br> • Investigated the similarity of the ML models results. <br> • Vibration signal analysis was performed while studying and extracting the pattern-behavior of bearings. <br> • CRA can recommend a fault with 93% accuracy. | [99] |

**Table 1.** *Cont.*

| ML Techniques | ML Cat. | Equipment/ System | Device Used for Data Acquisition | PdM Data Description | Data Size | Data Type | Key-Findings | Ref |
|---|---|---|---|---|---|---|---|---|
| SVM, MLP-ANN | C | Electrical power systems | - | Partial Discharge samples and noise samples dataset | 100 thousand datasets | Synthetic data & Real data | • Parallelism and pipelining methods can give a significant maximization of sample rates.<br>• The method provides better performance compared to other classifiers.<br>• Use low number of self-configuration capabilities and configuration parameters.<br>• The technique can deal with uneven dimensional input spaces.<br>• The model can process with simple evaluation function. | [100] |
| SVM, k-NN, MC | C | Tungsten filament - Ion implantation | Ion-implantation tool | Maintenance cycles dataset | N = 33 R2F maintenance cycles with 31 variables and 3671 batches | Real data | • Approach for integral faults type.<br>• Can be used in high-dimensional and censored data problems.<br>• Can be applied to any maintenance problems as R2F data is available.<br>• The model can be applied in health factor indicator.<br>• Shows better performance than other PdM classical methods and SVM.<br>• MC-PdM - k-nn outperforms PvM approaches.<br>• SVM offers better performance than k-nn. | [14] |

**Table 1.** *Cont.*

| ML Techniques | ML Cat. | Equipment/ System | Device Used for Data Acquisition | PdM Data Description | Data Size | Data Type | Key-Findings | Ref |
|---|---|---|---|---|---|---|---|---|
| Dynamic regression | R | Bearing | PRONOSTIA platform | Vibration measurements data | 2560 samples amplitude of the vibration signals | Real data | • Method for rolling element bearing.<br>• Predicts the bearing health and its URL using regression models.<br>• Used dimensionless quantity as bearing health indicator.<br>• ABT was used to determine TSP.<br>• Validated and tested on PRONOSTIA dataset.<br>• Achieved a reasonable result compared to the existing techniques. | [101] |
| GA-ANN, SVM | C | Rotating machine-Gear box | Acoustic emission and vibration sensors | Acoustic emission and vibration signals | (16 time-domain and 6 frequency-domain | Real data | • Investigated an early detection of the potential failures of rotating machine.<br>• Early misalignment detection was very hard using frequency analysis technique.<br>• The feature analysis method can detect a growth in fault. | [102] |
| SMDP | - | Ram feed-Boring machine | - | Feed straightness, positioning and hydrostatic pressure units | 18 degradation histories and 16 failure histories | Real data | • Developed a unit-level and system-level maintenance of a boring machine.<br>• Achieved low maintenance cost in comparison to other techniques.<br>• The model can be generalized to a wide variety of systems with a particular failure mode. | [103] |
| HC, k-means, PCA, model-based and Fuzzy C-Means clustering | Clustering | Exhaust fan | Vibration monitor sensor | Vibration data | The vibration was collected at every 240 min for 12 days with 41 observations | Real data | • Early fault detection of exhaust fan using several ML algorithms.<br>• $T^2$ statistic method is the best for faults detection only.<br>• Clustering algorithms is the best for fault detection under different levels.<br>• PAC produced better results compared to model-based techniques. | [12] |

**Table 1.** *Cont.*

| ML Techniques | ML Cat. | Equipment/ System | Device Used for Data Acquisition | PdM Data Description | Data Size | Data Type | Key-Findings | Ref |
|---|---|---|---|---|---|---|---|---|
| k-means | Clustering | Laser melting | Temperature & pressure sensors | Laser melting machine sensor data | 206 manufacturing processes data with 3D- matrix ($7 \times 3 \times 206$) | Real data | • Analyzing and visualizing offline-data from several sources. <br> • The clusters were identified by utilizing three sensors. <br> • Three faulty and normal operation stages were identified by the sensors. <br> • Implemented a CMS that permits machine tools for PdM solutions. | [104] |
| k-means, -DL, RNN, | C | - | Vibration sensors | Vibration data and categorical metadata. | 51 vibration sensors for 2.5 years | Real data | • Sequential data with categorical inputs were used in machine health predictions. <br> • Validated using an ablation study. <br> • Can be used without making any assumptions to the data on any dataset. | [105] |
| PCA, k-means | C, Clustering | Oil-immersed power transformer | - | Dissolved gas concentrations data | 46-observations with 6-variables | Real data | • A study for automatically extracting classes in dissolved gases. <br> • Class interpretation was according to the information obtained for each gas and TDCG. <br> • Permits to identify the major working periods of the power transformer. | [106] |
| FURIA, -ANN, SVM, RF, BayesNet, LogitBoost | C | Gas turbine | - | Big dataset of gas turbine | 11,934 instances with 16 independent variables for 49 measurements | Synthetic data | • Big data analytics maintenance optimizing through CBM. <br> • The FURIA model produces better decision boundaries. <br> • Simulation dataset of a sophisticated gas turbine propulsion plant is used. | [107] |

**Table 1.** *Cont.*

| ML Techniques | ML Cat. | Equipment/ System | Device Used for Data Acquisition | PdM Data Description | Data Size | Data Type | Key-Findings | Ref |
|---|---|---|---|---|---|---|---|---|
| DL | C | CNC machine | Accelerometer sensor | Vibrational data | 10,000 samples with 1024 data points for 288 days | Real data | • An early fault detection DL model under time varying conditions for CNC machine. • Developed from vibration signals to select impulse responses. • Experiment show that the proposed technique is not affected by time-varying conditions. • The model has the potential to detect an early fault in manufacturing. • The method can significantly detect the machine tool health status. | [108] |
| DT, ANN, RF, GNB, BNB | C | Anode production of industrial equipment | Process sensor | Process sensor data within the period of operation | 29 features and 96 faults with 4,852,153 data points | Real data | • PdM technique for fault forecast in real time before occurrence of an industrial equipment. • Predicts faults in industrial equipment 5–10 min before it occurs. | [109] |
| LR, DT, SVM, RF, kNN, K-Means, GBM, AdaBoost | - | Turbofan engine | Sensor run-to-failure measurements | Repository dataset of NASA for turbofan-engine | Dataset collected from 100 to 250 engines, each engine with 21 sensor values. | Real data | • A comparative study for ML techniques in predicting RUL of aircraft-turbofan engine. • The models can be applied in predicting faults before their occurrence. • Developed the ML models using NASA prognostics data of turbofan-engine. • Tested and validated, results obtained have shown a promising result. | [21] |
| MGGP | C | Metal lathe machine | Accelerometer and noise level meter | Vibration and acoustic signals data | 42 data samples | Real data | • Developed for probable faults detection of metal lathe machine using vibro-acoustic condition monitoring. • Proposed MGGP framework based on it is new measure of complexity. • The model outperforms the classical MGGP models. | [110] |

**Table 1.** *Cont.*

| ML Techniques | ML Cat. | Equipment/ System | Device Used for Data Acquisition | PdM Data Description | Data Size | Data Type | Key-Findings | Ref |
|---|---|---|---|---|---|---|---|---|
| RLS, SVM | R | LM 2500 - Gas turbine | Sophisticated simulator of a gas turbine | CODLAG propulsion plant data | $9 \times 51$ experiments | Synthetic data | • Techniques for the degradation forecasting of propulsion plant. <br> • Tested using realistic and sophisticated simulator of a gas-turbine. <br> • Effective solution in real application of maritime for CBM purposes. | [111] |
| HC, K-medoids, K-means | Clustering | Semiconductor manufacturing-CVD process | CVD process | Sensors dataset | 80-sensors for 6-months: 6,912,000 data points | Real data | • Validated the model using CVD process sensor data. <br> • Outperforms the model with full sensors. <br> • OPTICS was found to be the best of all the 5 algorithms. <br> • The model can be applied in engineers making decision of CBM. | [13] |
| ARIMA, GB, RF and RNN | R, Clustering | Machine tool of CNC Machine | Machine tool signal recorder | Spindle load, piece number, and tool position | 2 years data. 1100 series and 550,000 pieces were stored. | Real data | • Developed for simulation of RUL in a machining phase based on a linear regression model. <br> • Obtained more precise findings to predict the RUL for comparison. <br> • Can be applicable to other processes in production. | [78] |
| GBM, K-nearest Neighbors, BNB, NN, DT | C | Engine equipped with a rotating shaft | MEMS triaxial accelerometer | Vibration, noise, pressure, temperature, humidity | 4000 acceleration values | Real data | • Maximum peak in the Naïve Bayes algorithm with a precision loss of 5%. <br> • Only NN algorithm behaves differently compared to other algorithms. <br> • Best results obtained with isolation forest algorithm by increasing the training time with 20%. | [112] |
| LR, DTR, RFR, SVR | R | CNC machine tool | Accelerometer, Dynamometer, AE sensor | Vibration, force, temperature, federate, cutting depth | 3 sensors utilized for data collection | Real data | • Hybrid method predicts value close to actual value with small error. <br> • Data driven method provides less accuracy. <br> • Hybrid approach driven by Digital Twin provides more precise predicted values, 6.27% at the end stage. | [79] |

**Table 1.** *Cont.*

| ML Techniques | ML Cat. | Equipment/ System | Device Used for Data Acquisition | PdM Data Description | Data Size | Data Type | Key-Findings | Ref |
|---|---|---|---|---|---|---|---|---|
| | | | | | | | • Approach verified by a case study of the CNCMT tool life prediction. | |
| K-means, PCA | Clustering | Machine motor | Sensor, PLC, Production monitoring system | Power, torque, vibration, temperature | 13 datasets for machine1, 9 datasets for machine 2 | Real data | • There are inliers within the dataset which are seen in low frequency. <br> • Observed that torque measurement deviates from normal range and achieves the peak value. So that, the spindle of Machine1 may not work within the normal range. <br> • There are many inliers found and they can give diagnostic information. | [113] |
| ANN, SVM | C, R | Building facilities | IoT sensor network | Temperature, pressure, flow rate | 300 datasets for condition prediction. | Real data | • BIM and IoT facilitated the implementation of predictive maintenance to improve the feasibility of FMM process. <br> • Consists of two layers: information layer and application layer. <br> • Four modules are applied for maintenance: (1) condition prediction, (2) maintenance planning, (3) condition monitoring, (4) condition assessment. | [30] |
| ANN | C | Packaging robot | Sensor | Vibration, temperature, humidity | 157 failure for 2 years. 8 input, 20 hidden and 4 output nodes | Real data | • MLP structure can cope with unplanned downtime occurrence. <br> • Can reduce the unplanned production downtime costs immensely. <br> • Theoretical and practical comparison of failures provided. | [114] |
| SVM, LR | C, R | Nuclear power plant | Temperature sensor, pressure sensor | Temperature, power, current, speed | - | Synthetic data | • ML algorithm to predict maintenance of nuclear infrastructure. <br> • Power consumption is monitored. <br> • Temperatures within electrical panels are captured. | [115] |

**Table 1.** *Cont.*

| ML Techniques | ML Cat. | Equipment/ System | Device Used for Data Acquisition | PdM Data Description | Data Size | Data Type | Key-Findings | Ref |
|---|---|---|---|---|---|---|---|---|
| FFNN, RF, LR | C, R | Oil analysis of gearbox | Data obtained from oil analysis company | Oil condition (loss of additives or contamination detection) | 26 features with 887,255 samples collected from 126,644 gearboxes | Synthetic data | • Demonstrated the potential use of RF as a diagnostic tool in PdM.<br>• The RF models managed to achieve high recall, but the precision was low.<br>• RF achieved for LEAK a mean accuracy of 0.9837.<br>• For OVHT, POIL, and OTHR, none of the classification models showed significant classification ability. | [116] |
| DT, RF, LR | C, R | Backblaze data center | SMART sensor | vibration | 232,662 recordings in 2018. | Real data | • Simulated the operation of several predictive maintenance systems.<br>• Method used for bearing vibration failure data.<br>• RF technique performs the best in terms of predictive accuracy. | [117] |
| DNN, KNN | C, R | Turbofan engine | Multi-sensor | temperature, rotation speed, pressure | 21 featured data sets. | Real data | • C-RE has proven to be a robust feature selection method.<br>• Identifies faults or occurrence of faults during the asset's life cycle. | [118] |
| NN, weighted NN, DT | C | Boiler and heat pump of (HVAC) system | Sensor in each boiler and IoT platform | Temperature, number of boilers, heating requests, and cycle | The data was collected for 16 months, 1000 appliances of ten different models. | Real data | • Model exhibited best performance when the LSTM2 with 3 hidden layers of 50 neurons and the LSTM3 with 1 hidden layer of 25 neurons.<br>• Models exhibited worse performance than other techniques.<br>• Weighted NNs has poor precision compared with the NN models. | [119] |
| GBM, RF, XGBoost, NN classifier | C | Woodworking industrial machines | Big Data provided by machine tool data log systems | Vibration, current, temperature | Dataset divided into two groups: a training set (70%) and testing set (30%) | Real data | • up to 98.9%, 99.6%, and 99.1% accuracy, recall, and precision respectively.<br>• PdM implemented to Big Data stream processing system.<br>• Screens log files and forecasts the computer's status every 24 h. | [120] |

**Table 1.** *Cont.*

| ML Techniques | ML Cat. | Equipment/ System | Device Used for Data Acquisition | PdM Data Description | Data Size | Data Type | Key-Findings | Ref |
|---|---|---|---|---|---|---|---|---|
| RNN, SVMi k-nearest neighbor | C | Switchgear, | Infrared Sensor, thermopile array sensor | Temperature, voltage, current | - | Real data | • The combination of novel sensor technology and ML methods.<br>• Provides predictive maintenance solutions for medium voltage switchgear. | [121] |
| RF | C | Aircraft | Aircraft advanced sensors | Pre-defined parameters, failure messages | More than 7 years' worth of data collected. | Real data | • Model has a precision of more than 70%.<br>• Model predicts aircraft failure that leads to component replacement.<br>• Predicts more than 50% of aircraft component replacement. | [122] |
| LR | R | Jet engine blades | Flight sensor | Temperature, stress, strain | - | Real data | • 87% or higher in life consumption prediction for 19 out 21 blade that had failed at visual inspection. | [123] |
| MLP, LR, GBT, SVM | R, C, Clustering | Wind turbine | Public dataset library, SCADA | Rotation, temperature, itch angle, wind speed | | Real data | • MLP produces the most promising model for predicting failures on the given dataset.<br>• MLP models is a good way of producing a model with a lower variance than the individual base models. | [124] |

CNC-MM: CNC milling machine; PF: Particle filter; LVDT: Linear variable differential transformer; RVDT: Rotary variable differential transformer; GAPF: Genetic algorithm particle filter; DBN: Dynamic Bayesian Network; FDD: Fault detection and diagnosis; BC: Bayesian classification; DT: Decision tree; GMB: Gradient boosting machines; DNN: Deep Neural Network; GA: Genetic Algorithms; LS-PMSM: Line start-permanent magnet synchronous motor; DT: Decision tree: NBC: Naïve Bayes classifier; LR: logistic regression; 1NNC: 1-nearest neighbor classifier; TFI: Time–frequency image; IRT: Infrared thermal; STFA-PD: Sparse time–frequency analysis method based on the first-order primal-dual algorithm; SMDP: Semi-Markov decision process; APU: Auxiliary power unit; TF-IDF: Term Frequency-Inverse Document Frequency; HC: Hierarchical clustering; PCA: Principle component analysis; GLM: Generalized linear model; GBM: Gradient boosting machine; FURIA: Fuzzy unordered rule induction algorithm; D: Deterioration; SLM: Selective laser melting; GNB: Gaussian Naïve Bayes; BNB: Bernoulli Naïve Bayes; RNN: Recurrent neural network; RUL: Remaining useful life; LDA: Linear discriminant analysis; HDD: Hard Disk Drive; LSTM: Long Short Term Memory; SAFE: Supervised Aggregative Feature Extraction; CRA: Collaborative Recommendation Approach; MLP: Multilayer Perceptron; LVD: Logged Vehicle Data; VSR: Volvo Service Records; MC: Multiple classifier; MGGP: multi-gene genetic programming; BN: Bayesian Network; TASMI: Thermal Treatment equipment; R2F: Run to failure; RLS: Regularized least square; REB: Rolling element bearings; MOM: Manufacturing operation measurement; CVD: Chemical vapor deposition; DBSCAN: Density based spatial clustering of applications with noise; OPTICS: Ordering points to identify the clustering structure; FE: Finite element: FL: Feature learning; TPR: True Positive Rate; FPR: False Positive Rate; ABT: Alarm bound technique; TSP: Time to start prediction; CMS: Condition Monitoring System; CODLAG: COmbined Diesel eLectric And Gas propulsion plant; DPM: Dynamic Predictive Maintenance.

*4.11. Commercial Platforms available for Machine Learning in Smart Manufacturing Industry 4.0*

Data Science and Machine Learning Platforms offer platforms for the development, implementation, and analysis of machine learning algorithms. Such systems integrate intelligent algorithms for decision taking with data, thereby enabling developers to build a business solution. Several platforms provide pre-constructed algorithms and simple workflows with functionality such as drag and drop modeling and visual interfaces that quickly link required data to the end solution, whereas others need further programming and coding skills. In addition to other machine learning applications, these algorithms have functionalities for image recognition, natural language processing, speech recognition, and recommendation systems [125]. The most common platforms are mentioned in Table 2.

**Table 2.** Most common commercial platforms for ML.

| Software | Remarks |
|---|---|
| TensorFlow | • Open source software library for numerical computation |
| IBM Watson Studio | • ML framework designed for an AI-powered company. |
| RapidMiner | • Combines the whole lifecycle of data science from data planning to machine learning to predictive algorithm implementation. |
| Google Cloud AI Platform | • Machine learning on any data, of any size. |
| Box Skills | • Create structure and extract insights from your data at scale. |
| Google Cloud AutoML | • Train high-quality models specific to their enterprise needs.<br>• Google's transfer learning and Neural Architecture Search technology. |
| SAS Enterprise Miner | • Streamlines the data mining process to develop models quickly.<br>• Find the patterns that matter most for processes. |
| MATLAB | • Programming, modeling, and simulation tool developed by MathWorks. |
| IBM Watson Machine Learning | • Use of existing data to create, train, and deploy machine learning and deep learning models.<br>• An automated, collaborative workflow to grow intelligent business. |
| Anaconda Enterprise | • Harness data science, machine learning, and AI. |
| Amazon SageMaker | • Quickly build, train, and deploy machine learning models at any scale. |
| IBM Decision Optimization | • Combines mathematical and AI techniques to help decision-making such as operational, tactical/strategic planning, and scheduling processes. |
| IBM Cloud Pak for Data | • Modernizes data collection and data analytics to infuse AI across their organization. |
| BigML | • Programmatic machine learning. |
| H2O | • Decrease fraud and money laundering risks.<br>• Improve product design, marketing and business innovation.<br>• Early disease detection, drug discovery, personalized medicine. |
| Oracle Data Science Cloud Service | • Train, deploy, and manage models on the Oracle Cloud. |
| Domino | • Develop and deploy predictive models more rapidly. |
| Deep Cognition | • Design, train, and deploy ML models without coding. |
| KNIME Analytics | • Open source data analytics, reporting and integration platform. |
| Qubole | • Provides a simple and secure data lake platform for ML, streaming, and ad-hoc analytics. |

## 5. Discussion and Conclusions

The literature review is categorized based on ML techniques, ML categories, equipment used, device used in data acquiring, description of the applied data, data size, and data type. Based on

performed comprehensive literature review, predictive maintenance continues to be an important method for improving efficiency in all kinds of environments where machines that wear down over time are involved. The possibilities of manufacturing and placing cheap, connected sensors will continue to increase with the rise of IoT. As the amount of data increases with the number of sensors, so will the possibilities of applying machine learning algorithms to perform predictive maintenance.

This paper presents a comprehensive review of ML techniques applied in PdM of industrial components. Recent applications within the timeframe of ten years (i.e., 2010–2020) for several ML algorithms were reviewed and presented. Finally, some discussions have been drawn based on the literature review performed.

It is observed that predictive maintenance has enormous market opportunities, and that machine learning is an innovative solution to predictive maintenance implementation. Yet, according to a PwC survey, only 11% of the companies surveyed have "realized" predictive maintenance based on ML [126]. There are some challenges implementing ML algorithms for PdM in I4.0 and those are identified in Table 3.

**Table 3.** Challenges in implementing ML for Industry 4.0 (I4.0).

| Challenges | Remarks | References |
|---|---|---|
| Identification of required data to collect | • Launch of connected machines. <br> • Unclear evidence of data that provide value. <br> • Unclear business goal and planning. | [33] |
| Getting required dataset | • Without input data, it is not possible to run ML algorithm. <br> • Much time and resources to establish ML solutions. <br> • Choosing wrong ML algorithm causes loss time and loss in cost. | [29] |
| Enhanced data science | • Determine an appropriate method of analyzing the data. <br> • Choosing a correct method of presenting the data-driven insights. | [126] |
| Security | • Safeguarding admission to critical equipment. <br> • Proactive approach to cybersecurity whilst protecting connected assets | [29,33,126] |

SVM, RF, and ANN, are the most used ML algorithms in reviewed literature. They have been successfully applied in several field of PdM applications. Some authors [10,30,46–48,75–77,107,114,118–121] focused on ANN ML algorithms. Some other authors [64–67,81,82,84–88,90,116,117] studied RF technique. In the last years, use of SVM technique has received attention from authors [14,21,48,56,66, 79,93–100,102,107,111,115,121,124]. Some authors [21,56,82,112,120,124] have given attention to GBM technique. Some authors [65,66,79,109,112,117] carried out studies by use of DT technique. However, it is observed that there is less consideration given to XGBoost technique and there are only a few studies found in the literature [62,81,120].

Based on the literature, it is observed that RF algorithm is the most extensively applied ML technique for PdM, as it has been applied on various industrial equipment, components, or systems, including rotating machineries, turbofan engines, rotor bar-LS-PMSM, aircrafts, production lines, semiconductors, industrial pumps, cutting machines, refrigeration systems of supermarket, hard drive disk (HDD), wind turbine, vending machines, and many others. However, authors usually focused on CNC Machines [46,78,79], wind turbine [10,89,99,124], aircraft [67,94,97,122,123], and semiconductor [13,81–83].

Most of the data type used is real data; very few studies applied simulated or synthetic data while developing the ML algorithms. Public data applied in developing ML algorithm PdM models

include Bosch data, SECOM data, Repository dataset of NASA for turbofan-engine, CMAPSS dataset of aircrafts, CMAPSS NASA simulation dataset of engine degradation.

- Vibration signals acquired using accelerometer are the most used data.
- The most applied ML category is classification.

According to existing research, a couple of limitations are highlighted. The limitations are the following: (1) Although the classifiers have presented excellent accuracy in the distinction between states, they are required to be trained with a complete dataset of all the faults. (2) Algorithms are selected base on developer's experience and this situation can have influence on variable of prediction results. (3) Carrying out a study with a single prediction method may not present excellent results. Therefore, application of other methods to provide comprehensive results between methods can give better understanding about the study. (4) Cross validation for models can be unsuccessful due to lack of RAM memory.

Moreover, some of the works conducted by this research employ regular machine learning methods without parameter tuning. Perhaps this is due to the fact that PdM is a new subject for industry experts and is beginning to be explored. It is also important to point out that it is appropriate to have the R2F and PvM strategies already applied in its process to collect data for PdM modeling in order to obtain good results of a PdM strategy in a plant. Based on that data, designing and validating a PdM strategy becomes feasible. During this study, it was noted that there is incremental application of machine learning techniques to develop PdM applications. Integrating PdM and machine learning in some applications provides cost reduction. However, the incorporation of PdM techniques with the new sensor technologies can be seen as avoiding unnecessary replacement of equipment, saving costs and improving process safety, availability, and performance.

This paper presents a comprehensive review of ML techniques in PdM by identifying the most used ML techniques, industrial areas where ML is applied, and utilized data type for ML applications, and proposes a way forward, and provides a foundation for further research. Below, remarks for further researches are made to motivate authors and practitioners.

- Extraction of real time data using intelligent data acquisition system can help to automate predictive maintenance.
- Combination of more than one ML models can provide better prediction compared to use of individual model.
- ML model implementations based on cloud can be further studied.
- Classification and Anomaly Detection algorithms can be combined to maintain precision of classification models without losing Anomaly detection advantages. By this way, PdM can be applied to equipment or system which does not have large dataset.

**Author Contributions:** Z.M.Ç. was responsible for data curation, resources, visualization. B.S. was responsible for investigation. Q.Z. and O.K. were responsible for supervision, methodology. B.S., Q.Z., O.K., M.A., A.A.N. and Z.M.Ç. were responsible for writing—original draft and writing—review and editing. B.S., Q.Z. and O.K. were responsible for project administration. All authors have read and agreed to the published version of the manuscript.

**Funding:** This research received no external funding.

**Conflicts of Interest:** The authors declare no conflict of interest.

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
