# Peer review of "Machine Learning in Predictive Maintenance towards Sustainable Smart Manufacturing in Industry 4.0"

_sustainability, doi:10.3390/su12198211_

Round 1
Reviewer 1 Report
The manuscript includes a comprehensive literature review on the use of machine learning in predictive maintenance for Industry 4.0. It can be improved if the authors address the following points:
- The manuscript includes many single-sentence paragraphs. A paragraph with only one sentence is not appropriate and it hampers the logical flow. I recommend making every paragraph include at least two sentences.
- Introduction section is a bit wordy and inconsistent. It would be better if this section is more concise. Paragraphs in lines 72-93 start with incomplete sentences. It seems colon (:) should be replaced with comma (,) to make them complete sentences.
- Same issues continue throughout the manuscript. "PdM and ML Techniques" section also needs to be more concise and redundancies should be reduced.
- In Table 1, data type needs to be clearly explained.
- In "Discussion and Conclusion" section, it looks like a collection of observations and findings from the literature review without any congruent or overarching conclusion. This section needs to be written so that it can provide a more insightful logically-connected message to readers rather than disconnected findings.
Author Response
Response To Reviewers
Manuscript ID: sustainability-917376
Machine Learning in Predictive Maintenance towards Sustainable Smart Manufacturing in Industry 4.0
Zeki Murat Çınar, Abubakar Abdussalam Nuhu, Qasim Zeeshan, Orhan Korhan, Mohammed Asmael and Babak Safaei
Dear Editor:
We wish to thank you for the timely processing of our article in “Sustainability” and the reviewers for their constructive comments, which have helped us to clarify a number of issues in our manuscript. We have attended to all the raised points/concerns in this revised manuscript. Given below is our response and the suggested revisions to the article, as highlighted in the manuscript.
Sincerely yours,
Dr. Babak Safaei
PhD (Tsinghua University)
Assistant Professor of Mechanical Engineering
Eastern Mediterranean University
Dated: 10.09.2020
Reviewer’s comments:
Reviewer #1
The manuscript includes a comprehensive literature review on the use of machine learning in predictive maintenance for Industry 4.0. It can be improved if the authors address the following points:
Response:
We are thankful to the esteemed reviewer for the appreciation of our manuscript and grateful for the insightful comments. It indeed helped us in improving the quality of our manuscript. The manuscript has been revised and improved as per the suggestions and instructions of the reviewer. Thank you.
- The manuscript includes many single-sentence paragraphs. A paragraph with only one sentence is not appropriate and it hampers the logical flow. I recommend making every paragraph include at least two sentences.
Response:
Line number 194 (page 6) à Paragraph content changed according to reviewer’s recommendation. Now it has 2 sentences. As this is the ethical statement of the paper, it must stay as a separate paragraph.
“Ethical/Legal permission is not required for this study. This rstudy complies with research and publication ethics in obtaining all kinds of data, images.”
Line 69 (Page 2)à paragraph has only one sentence. It is connected with previous paragraph.
“…. Moreover, fault detection is one of the critical components of predictive maintenance, it is very much needed for industries to detect faults at very early stage [12]. Techniques for maintenance policies can be categorized into the following main classifications [13–17],”
Line 93 (page 3à paragraph has only one sentence. It is connected with previous paragraph.
“It is required that any maintenance strategy ought to minimize equipment failure rates, must improve equipment condition, should prolong the life of the equipment, and reduce the maintenance costs. An overview for the maintenance classifications is shown in Figure 1. PdM turned out to be one of the most promising strategies amongst other strategies of maintenance that has the ability of achieving those characteristics [19], thus the strategy has been applied recently in many fields of studies. PdM captivates the attention of the industries, hence it has been applied in the era of I4.0 due to it is capability of optimizing the use and management of assets [1,20]. “
Line 102 (page 4) à paragraph has only one sentence. It is connected with previous paragraph.
“ML, within the contexts of AI (Figure 1), lately, has appeared to be one of the most powerful tools that can be applied in several applications to develop intelligent predictive algorithms. It has been developed into a wide field of research over the past decades. ML can be defined as a technology by which the outcomes can be forecasted based on a model prepared and trained on past or historical input data and its output behavior [22]. According to Samuel, A.L. [23], ML mainly means, if computers are allowed to solve without specifically being programmed in doing so. ML approaches are known to have tremendous advantages, as they have the ability in handling multivariate, high dimensional data, and can extract hidden relationships within data in complex, dynamic and chaotic environments [1,24,25]. However, depending on the ML approach chosen, the performance and advantages might differ. As of today, ML techniques have been widely applied in several areas of manufacturing (such as, maintenance, optimization, troubleshooting, and control) [24]. Consequently, this paper aims to provide the recent advancements of ML techniques applied to PdM from an ample perspective. Predominantly, this ample review uses Scopus database while acquiring and identifying the articles used. From a comprehensive perspective, this paper aims to pinpoint and categorize based on ML technique considered, ML category, equipment used, device used in data acquiring, applied data description, data size and data type.”
- Introduction section is a bit wordy and inconsistent. It would be better if this section is more concise. Paragraphs in lines 72-93 start with incomplete sentences. It seems colon (:) should be replaced with comma (,) to make them complete sentences.
Response:
Page 3à below paragraph removed.
“In general, equipment maintenance and machine health prediction are two important terms for industries and they do not only affect the equipment performance (efficiency) and operation time, they have also significant impact on downtime, labor costs, operation and maintenance plan. By the identification of faults of industrial equipment, production can be controlled in a more efficient way that can avoid any unexpected shutdown in the manufacturing processes [11].”
Line 72 (page 2) à Semicolon is replaced with coma as requested.
“ Techniques for maintenance policies can be categorized into the following main classifications [13–17],”
Line 79-93 à Incomplete sentences are fixed as below.
“1. (R2F) Run 2 Failure: also known as corrective maintenance or unplanned maintenance. It is the simplest amongst maintenance techniques which is performed only when the equipment has failed. It may lead to high equipment downtime and a high risk of secondary faults. Thus, create very large number of defective products in production.”
“2. Preventive Maintenance (PvM): also known as scheduled maintenance or Time-based maintenance (TBM). PvM refers periodically performed maintenance based on a planned schedule in order to anticipate the failures. It sometimes leads to unnecessary maintenance which increase the operating costs. The main aim here is to improve the efficiency of the equipment by minimizing the failures in production [18].”
“3. Condition-based Maintenance (CBM): this method of maintenance is based on a constant machine or equipment monitoring or their process health that can be carried out on only when they are actually necessary. The maintenance actions can only be carried out when the actions on the process are taken after one or more conditions of degradation of the process. CBM usually cannot be planned in advance.”
Page 7 à Paragraph is removed to make introduction section more concise as requested.
“PdM techniques can be categorized into the following three main classifications, they are the approaches that capable of monitoring the equipment conditions for diagnostic and prognostic applications [21]: 1) statistical, 2) artificial intelligence (AI), and 3) model-based methodologies. Whereas, according to [4], PdM methodologies are categorized as: 1) model-based prognosis, 2) knowledge-based prognosis, and 3) data-driven prognosis. However, more or less their study serves the same purpose. Artificial intelligence techniques (such as Machine Learning techniques and Deep Learning) are the most applied recently in the field of PdM applications. There is a need for mathematical background for statistical approaches, and model-based require mechanistic knowledge and theory of the equipment to be monitored [1].”
- Same issues continue throughout the manuscript. "PdM and ML Techniques" section also needs to be more concise and redundancies should be reduced.
Response:
Line 126 (page 3) à redundancies are reduced and incomplete sentences fixed.
“Currently, the PHM system has become a safe-fire method for maintaining the safety status of equipment (e.g. defect detection and Remaining Useful Life (RUL)). It is accomplished by the systematic use of the current testing findings in AI technology and IT technology. [4]. Additionally, PdM cannot only provide reduction in the costs of the maintenance, it can also prolong the RUL [26]. The incipient issues that may lead to disastrous failures can be correctly forecast and appropriate steps can be set in order to avoid these failures on the basis of the prediction outcomes [4]. Nevertheless, at any appropriate time an industrial equipment can be replaced or repaired before the fault happens, thus might restore the original condition of the equipment or system after each and every completed maintenance. Moreover, the equipment, component, system or a machine health status can be obtained at any instances. Also, their failures can be predicted in order to achieve a non-zero downtime performance [27]. PdM mainly focuses to utilize predictive info in order to accurately schedule the future maintenance operations [21].”
Line 138 (page 4) à incomplete sentences fixed.
“Aim of PdM is not only to collect process data and it is parameters, but also to collect the physical health aspects of the equipment, machine, or component (such as pressure, vibration, temperature, viscosity, acoustics, viscosity, flow rate data) and many as such. At the same time, this information collected are now widely used for fault identification, early fault detection, equipment health assessment, and predicting the future state of the equipment [12].”
Line 162 (page 5) à redundancies are reduced and incomplete sentences fixed.
“In unsupervised machine learning, there is no feedback from an external teacher or knowledgeable expert [24]. Based on the existing data, the algorithm identifies the clusters. The main aim here in supervised learning is determining the unknown classes of items by clustering [36]. whereas, classification is for supervised learning. Unsupervised ML basically defines any ML method that attempts to learn ‘structure in the absence of either an identified output (like supervised ML) or feedback (like RL) [24]. “
Line 182 (page 6) à incomplete sentences fixed.
“There are several available supervised machine learning algorithms as few can be seen from Error! Reference source not found.. Each of these algorithms has its own specific advantages as well as limitations regarding the application (either PdM or manufacturing). Selecting the most appropriate and suitable ML algorithm can be a major challenge for the requirements of the PdM problem.”
- In Table 1, data type needs to be clearly explained.
Response:
Table 1à In table 1 data types are clearly stated such as real data or Synthetic data in each row (in blue).
A paragraph added to line 186 (page 6)
“It is also important to getting good at applied machine learning is practicing on lots of different datasets. Therefore, each problem requires different subtly, different data preparation and modelling methods. Datasets are classified into seven categories that are multivariate, sequential text, time series, sequential, univariate, text and domain theory. However, this paper classifies datasets into two categories. One of them is real datasets that are any production data obtained from real production processes and applicable to ML. Other one is synthetic datasets that are. Any kind of production data applicable to ML but they are simulated data rather than direct measurement in the production. “
Also, below paragraph give brief discussion about data types. (line 62, page 33)
“Most of the data type used is real data, very few studies applied simulated or synthetic data while developing the ML algorithms. Public data applied in developing ML algorithm PdM models include Bosch data, SECOM data, Repository dataset of NASA for turbofan-engine, CMAPSS dataset of aircrafts, CMAPSS NASA simulation dataset of engine degradation;”
- In "Discussion and Conclusion" section, it looks like a collection of observations and findings from the literature review without any congruent or overarching conclusion. This section needs to be written so that it can provide a more insightful logically-connected message to readers rather than disconnected findings.
Response:
Line 620 (page 33) à In order to give an insightful message to readers, below paragraph is added in conclusion.
“Moreover, some of the works conducted by this research employ regular machine learning methods without parameter tuning. Perhaps this is due to the fact that PdM is a new subject for industry experts and is beginning to be explored. It is also important to point out that it is appropriate to have the R2F and PvM strategies already applied in its process to collect data for PdM modeling in order to obtain good results of a PdM strategy in a plant. Based on that data, designing and validating a PdM strategy becomes feasible. During this study, it was noted that there is incremental application of machine learning techniques to develop PdM applications. Integrating PdM and machine learning in some applications provides cost reduction. However, the incorporation of PdM techniques with the new sensor technologies can be seen as avoiding unnecessary replacement of equipment, saving costs and improving process safety, availability and performance.”
Best Regards

Reviewer 2 Report
Good introduction with adequate definitions + literature review.
Also, very interesting review of ML algorhytms in part 4.
Table 1 should be formatted in a better style (no need for bullets, etc..)
To sum up, this review article is a good quality, it analyses over 800 articles.
Also good list of references...
Author Response
Response To Reviewers
Manuscript ID: sustainability-917376
Machine Learning in Predictive Maintenance towards Sustainable Smart Manufacturing in Industry 4.0
Zeki Murat Çınar, Abubakar Abdussalam Nuhu, Qasim Zeeshan, Orhan Korhan, Mohammed Asmael and Babak Safaei
Dear Editor:
We wish to thank you for the timely processing of our article in “Sustainability” and the reviewers for their constructive comments, which have helped us to clarify a number of issues in our manuscript. We have attended to all the raised points/concerns in this revised manuscript. Given below is our response and the suggested revisions to the article, as highlighted in the manuscript.
Sincerely yours,
Dr. Babak Safaei
PhD (Tsinghua University)
Assistant Professor of Mechanical Engineering
Eastern Mediterranean University
Dated: 10.09.2020
Reviewer’s comments:
Reviewer #2
Good introduction with adequate definitions + literature review. Also, very interesting review of ML algorithms in part 4. To sum up, this review article is a good quality, it analyses over 800 articles. Also, good list of references...
Response:
We are thankful to the esteemed reviewer for the appreciation of our manuscript and grateful for the insightful comments. It indeed helped us in improving the quality of our manuscript. We have tried to incorporate the changes as proposed by the reviewer. Thank you.
- Table 1 should be formatted in a better style (no need for bullets, etc..)
Response:
Table 1 (page 20)à bullets are used to highlight remarks the key finding of the research in column. If bullets are removed, then table become more confusing and unclear. However, Table format is corrected, and shaped in a better style to provide better understanding.
Best Regards,

Reviewer 3 Report
This is an extraordinary piece of research with much valuable data. There are only four comments that I propose for consideration by the authors and by the editors.
- The research question does not appear till late in the Introduction section. This is a long section and the objective of the research is unclear in the absence of the stated objective.
- The Introduction section is repetitive and needs to be edited
- The inclusion of Table 1 is an issue for the editor. In usual circumstances, this would be summarised and integrated into the earlier discussions about the MI options
- You say that according to a PwC report, only 11% of companies are adopting these preventative mainitenance opportunities, and provide [33] as the reference. This report is not at [33] and could not be found by a google search
Author Response
Response To Reviewers
Manuscript ID: sustainability-917376
Machine Learning in Predictive Maintenance towards Sustainable Smart Manufacturing in Industry 4.0
Zeki Murat Çınar, Abubakar Abdussalam Nuhu, Qasim Zeeshan, Orhan Korhan, Mohammed Asmael and Babak Safaei
Dear Editor:
We wish to thank you for the timely processing of our article in “Sustainability” and the reviewers for their constructive comments, which have helped us to clarify a number of issues in our manuscript. We have attended to all the raised points/concerns in this revised manuscript. Given below is our response and the suggested revisions to the article, as highlighted in the manuscript.
Sincerely yours,
Dr. Babak Safaei
PhD (Tsinghua University)
Assistant Professor of Mechanical Engineering
Eastern Mediterranean University
Dated: 10.09.2020
Reviewer’s comments:
Reviewer #3
This is an extraordinary piece of research with much valuable data. There are only four comments that I propose for consideration by the authors and by the editors.
Response:
We are thankful to the esteemed reviewer for the appreciation of our manuscript and grateful for the insightful comments. It indeed helped us in improving the quality of our manuscript. The manuscript has been revised and improved as per the suggestions and instructions of the reviewer. Thank you.
- The research question does not appear till late in the Introduction section. This is a long section and the objective of the research is unclear in the absence of the stated objective.
Response:
Line 47 (page 2) à a paragraph added to introduction to highlight gap and ho our study fills the gap.
“However, it is very cruel to select appropriate ML techniques, type of data, data size and equipment to apply ML in industrial systems. Selection of inappropriate PdM technique, data set and data size may cause time loss and infeasible maintenance scheduling. Therefore, this study aimed to present a comprehensive literature review to discover existing studies and ML applications. Thus, helps researchers and practitioners to select appropriate ML techniques, data size and data type to obtain a feasible ML application.”
- The Introduction section is repetitive and needs to be edited
Response:
Line 66 (page 2) à repetitive paragraphs removed and summary paragraph added as below.
Added/modified sentence
“ML applications provides some advantages which includes; maintenance cost reduction, repair stop reduction, machine fault reduction, spare-part life increases and inventory reduction, operator safety enhancement, increased production, repair verification, increases overall profit, and many more. Also, these advantages have a tremendous and a strong bond with the procedures of maintenance [1,8–10]. “
Removed sentence:
“In general, equipment maintenance and machine health prediction are two important terms for industries and they do not only affect the equipment performance (efficiency) and operation time, they have also significant impact on downtime, labor costs, operation and maintenance plan. By the identification of faults of industrial equipment, production can be controlled in a more efficient way that can avoid any unexpected shutdown in the manufacturing processes [11].”
- The inclusion of Table 1 is an issue for the editor. In usual circumstances, this would be summarized and integrated into the earlier discussions about the MI options
Response:
Line Table 1 (page 20) à Sentences in table is summarized. Modified sections are highlighted in red in the paper. In this table we provide key finding for each paper and provided an extensive discussion in discussion section. Integration of this table to previous sections causes paper flow problems. Therefore, it provides better flow if we keep table in same location. We have also thought to separate the table, but keeping all knowledge in a single table may be better for researchers who would like to compare studies in terms of data sets, data type and key finding.
- You say that according to a PwC report, only 11% of companies are adopting these preventative maintenance opportunities, and provide [33] as the reference. This report is not at [33] and could not be found by a google search.
Response:
Line 585 (page 32) à Below reference fixed.
“It is observed that predictive maintenance has enormous market opportunities, and that machine learning is an innovative solution to predictive maintenance implementation. Yet, according to a PwC survey, only 11% of the companies surveyed have 'realized' predictive maintenance based on ML [126].”
Best Regards,

Round 2
Reviewer 3 Report
As was mentioned in my first review, there are features of this report that are for the editor to decide. That remains the case. The essence of the issue is whether this report in fact addresses two separate questions that ideally, at least in my opinion are best presented in two reports, rather than combined into one as is the case here. While thorough and technically creditable, the report speaks to the issues of the theory, and as well and in geat detail the product options and selection criteria. The audience for each are potentially different although there is a clear overlap. As separate reports, the interest and readabilitiy of the report would be greatly enhanced.
My recommendation of a major revision is for the reason of the consideration that this report is better presented as two reports. This is for the editor to decide, or maybe your choice.
Regarding the other issues that were raised in my first review, these have all been satisfacotiy addressed.
For your consideration, and that of the editor.
Author Response
Response To Reviewers – ROUND 2
Manuscript ID: sustainability-917376
Machine Learning in Predictive Maintenance towards Sustainable Smart Manufacturing in Industry 4.0
Zeki Murat Çınar, Abu Bakr Abdussalam NUHU, Qasim Zeeshan, Orhan Korhan, Mohammed Asmael and Babak Safaei
Reviewer #3
As was mentioned in my first review, there are features of this report that are for the editor to decide. That remains the case. The essence of the issue is whether this report in fact addresses two separate questions that ideally, at least in my opinion are best presented in two reports, rather than combined into one as is the case here. While thorough and technically creditable, the report speaks to the issues of the theory, and as well and in geat detail the product options and selection criteria. The audience for each are potentially different although there is a clear overlap. As separate reports, the interest and readabilitiy of the report would be greatly enhanced.
My recommendation of a major revision is for the reason of the consideration that this report is better presented as two reports. This is for the editor to decide, or maybe your choice.
Regarding the other issues that were raised in my first review, these have all been satisfacotiy addressed.
For your consideration, and that of the editor.
Response:
We are thankful to the esteemed reviewer for the appreciation of our manuscript and grateful for the insightful comments. It indeed helped us in improving the quality of our manuscript. The manuscript has been revised and improved as per the suggestions and instructions of the reviewer in Round 1. However, please see our response below for your valuable comments in Round 2.
Thank you.
Machine Learning Algorithms, Predictive Maintenance and Smart Manufacturing have been discussed in detail separately by many researchers recently. We have done an effort to provide a comprehensive review of the recent advancements of ML techniques widely applied to PdM for smart manufacturing in I4.0 by classifying the research according to the ML algorithms, ML category, machinery and equipment used, device used in data acquisition, classification of data, size and type, and highlighting the key contributions of the researchers, thus offering the guidelines and foundation for further research.
We understand and appreciate the opinion of the esteemed reviewer. There is a scope of a particular research and we sincerely believe that it is not possible to split this manuscript into 2 articles as it will lose its scope, structure and quality. We strongly believe that our manuscript qualifies in the quality and quantity in its present form to be included in your esteemed journal.
We therefore, humbly request you to please accept our manuscript in its present form.
Thanking you in anticipation.
Sincerely yours,
Dr. Babak Safaei
PhD (Tsinghua University)
Assistant Professor of Mechanical Engineering
Eastern Mediterranean University
Dated: 20.09.2020
